# PARAMETER-EFFICIENT SUBSPACE OPTIMIZATION FOR LLM FINE-TUNING

## ABSTRACT

This paper develops a new perspective on parameter-efficient fine-tuning for LLMs, inspired by the classical theory of subspace minimization. We introduce a unifying framework, **P**arameter-**E**fficient **S**ubspace **O**ptimization (**PESO**), which not only recovers many existing methods such as LoRA but also bridges them with the principled algorithmic and theoretical foundations of subspace optimization. This connection highlights a natural "exploration–exploitation" view of subspace methods, guiding the design of new algorithms that achieve strong convergence performance while still preserving memory efficiency. Importantly, our framework establishes the convergence in the full-parameter space, resolving a critical gap of LoRA variants where low-rank updates lack such guarantees. We further instantiate the framework into a practical algorithm named PESO-LoRA, based on LoRA-type parameterization. Our algorithm achieves notable improvements over existing methods on standard benchmarks.

## 1 INTRODUCTION

Pre-training and fine-tuning deep neural networks are the cornerstones of modern AI, powering the success of large-scale foundation models such as Large Language Models (LLMs) (Brown et al., 2020). At their core, both procedures reduce to solving a high-dimensional optimization problem over weight matrices:

$$\Delta W^* := \arg\min_{\Delta W} \ \ell\big(W_0 + \Delta W\big), \tag{1}$$

where $\ell(\cdot)$ is the loss function, $W_0$ is the initialization, and $\Delta W$ the increment. In practice, (1) is typically solved by first-order methods such as Adam (Kingma & Ba, 2014) and AdamW (Loshchilov & Hutter, 2017), which are the workhorses of large-scale training. However, these methods require storing additional optimizer states (e.g., momentum and velocity), and for LLMs this overhead places enormous pressure on memory resources, making parameter-efficient strategies appealing.

In the realm of fine-tuning, we often have limited labeled data for a downstream task but still wish to adapt the pretrained weights effectively and efficiently. Therefore, updating the entire parameter set is memory-intensive. This motivates the study of Parameter-Efficient Fine-Tuning (PEFT) methods (Han et al., 2024; Houlsby et al., 2019; Hu et al., 2022), where optimization is restricted to a smaller set of parameters initialized from pretrained weights. In other words, $W_0$ denotes weights obtained from a large-scale pretraining phase, and $\Delta W$ is not updated freely but instead follows an efficient parameterization that constrains the search space.

A popular PEFT method is low-rank adaptation (LoRA, Hu et al. (2022)), where matrices in $\Delta W$ are expressed as the product of two low-rank factors. LoRA has shown strong empirical success, reducing memory costs while achieving competitive downstream performance. However, it suffers from two key limitations: 1) performance often lags behind full-parameter fine-tuning (Figure 1, left: MetaMathQA); 2) theoretical guarantees are limited, with convergence typically shown only for the low-rank factors (Figure 1, middle: a synthetic example illustrating LoRA's potentially unbounded loss gap). To address these issues, many LoRA variants (Hayou et al., 2024; Wang et al., 2024a;b; Zhang et al., 2023; 2025b) have been proposed, yet they largely inherit the same shortcomings and leave the following fundamental question open:

*Can we design fine-tuning methods that maintain the practical performance of LoRA while still enjoying the convergence and optimality of full-parameter fine-tuning?*

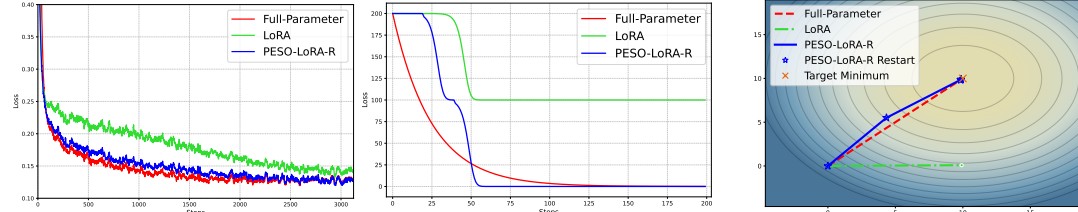

Figure 1: Comparison of full-parameter tuning, LoRA, and our method (PESO-LoRA). Left: Meta-MathQA. Middle: synthetic example $\min_W \|W - M\|_F^2$ with $M = 10 \cdot \mathrm{diag}(1, \ldots, 1, 0, \ldots, 0)$ ($r+1$ ones); see Appendix A. Right: optimization trajectories. PESO-LoRA bridges the loss gap of LoRA while preserving memory and computation efficiency.

To address this question, we reveal an inherent connection between PEFT and the classical idea of **subspace minimization**, a long-standing nonlinear optimization strategy dating back to Conn et al. (1994); Cragg & Levy (1969). The central philosophy is to decompose a large-scale problem like (1) into **iterative, simpler** subproblems constrained to carefully chosen subspaces. This view resonates naturally with modern PEFT methods, which restrict updates to structured low-rank forms for better efficiency. Interestingly, subspace minimization historically received less attention in the optimization society, since full-parameter information were often affordable in traditional applications. However, it is especially well suited to LLM training, where *massive dimensionality* calls for *memory-efficient* methods.

Formally, we build on the notion of *intrinsic dimensionality* in LLM training (Aghajanyan et al., 2020; Li et al., 2018), recognized in (Hu et al., 2022) as the origin of LoRA: there exists a dimension-lifting map $\mathcal{M} : \mathbb{R}^d \rightarrow \mathbb{R}^{m \times n}$, with $d \ll m \times n$, such that the optimal solution $\Delta W^*$ of (1) satisfies

$$\Delta W^* \approx \mathcal{M}(\xi^*), \quad \xi^* := \arg\min_{\xi \in \mathbb{R}^d} \ell(W_0 + \mathcal{M}(\xi)). \tag{2}$$

Here, $d$ stands for the number of trainable parameters, and this characterization implies that it suffices to optimize within the reduced space defined by the image of $\mathcal{M}$ to approximate $\Delta W^*$. For clarity, we focus on a single weight matrix $\Delta W \in \mathbb{R}^{m \times n}$ (multi-layer extensions are straightforward) and represent $\xi$ as a $d$-dimensional vector. This is without loss of generality, since tensor parameters can always be flattened via vectorization into an isomorphic Euclidean space. For example, LoRA adopts the simple form $\mathcal{M}(A, B) = AB$ with $A \in \mathbb{R}^{m \times r}$, $B \in \mathbb{R}^{r \times n}$, and $d = (m+n)r$. However, it remains unclear whether such a simple $\mathcal{M}$ is sufficient to capture the complexity of LLM training dynamics.

Our framework approximates $\mathcal{M}$ adaptively through a *sequential subspace approximation*, providing a more effective capture of (2). We construct a sequence of maps $\{\mathcal{M}_k\}$, each with a simple representation and $\mathbb{R}^d \rightarrow \mathbb{R}^{m \times n}$,

$$\Delta W^* \approx \sum_k \mathcal{M}_k(\xi_k^*), \quad \xi_k^* := \arg\min_{\xi \in \mathbb{R}^d} \ell(W_0 + \sum_{i=1}^{k-1} \mathcal{M}_i(\xi_i^*) + \mathcal{M}_k(\xi)). \tag{3}$$

where each image of $\mathcal{M}_k$ approximates a subspace and $\xi_k^*$ is its low-dimensional coordinate. In essence, the complexity of $\mathcal{M}$ is captured by a sequence of piecewise-linear subspaces. This philosophy parallels classical approximation schemes in numerical analysis such as finite element methods (Bathe, 2006).

Guided by this perspective, we develop a principled framework for PEFT grounded in subspace minimization, named **P**arameter-**E**fficient **S**ubspace **O**ptimization (**PESO**). A key insight is to view the problem (3) through an **exploration–exploitation** lens: *exploration* designs new subspaces that capture full gradient information, while *exploitation* optimizes efficiently within the current subspace. This resolves LoRA's two central limitations: lack of full-parameter convergence guarantees and inefficiency from rigid low-rank parameterization; see Figure 1.

**Contributions**. Our contributions can be summarized at three levels. Although our focus is on PEFT, many of the ideas developed here naturally extend to pre-training.

**I. Perspective Level.** We introduce a novel framework PESO for memory-efficient training inspired by classical subspace minimization (Conn et al., 1994), unifying existing PEFT approaches such as LoRA variants (Hu et al., 2022; Wang et al., 2024a;b; Zhang et al., 2023; 2025b) and GaLore (Zhao et al., 2024). This framework allows us to explore the rich algorithmic techniques in subspace methods, providing systematic guidance to improve memory-efficient methods. In particular, we highlight two complementary directions: *exploration* of new subspaces through information from the full gradient, and *exploitation* of the current subspace via streaming SVD representations.

**II. Theoretical Level.** Our exploration mechanism, *full gradient restart*, enables the framework to effectively guide training dynamics. The resulting algorithm is, to our knowledge, the first method for LLM fine-tuning that combines the practical effectiveness of LoRA-style designs with a provable convergence to full-parameter optimality up to small errors, without requiring additional assumptions such as explicit low-rankness of the solution.

**III. Empirical Level.** Guided by our PESO framework, we show that two practical instantiations of our framework—PESO-LoRA-R and PESO-LoRA-T—achieve improved performance while preserving the memory efficiency of state-of-the-art PEFT methods across benchmarks such as GLUE, mathematical reasoning, code generation, and general instruction tuning.

**Related Work.** LoRA (Hu et al., 2022) is perhaps the most widely known PEFT method, and numerous variants have been proposed for better performance. For instance, LoRA+ (Hayou et al., 2024) introduces imbalanced learning rates; PiSSA (Meng et al., 2024) proposes an initialization from SVD of $W_0$; and AdaLoRA (Zhang et al., 2023) maintains an adaptive SVD-based low-rank representation. Other extensions focus on gradient scaling (Tastan et al., 2025; Zhang & Pilanci, 2024). More recent work leverages information from the full gradient: LoRA-GA (Wang et al., 2024a) and LoRA-Pro (Wang et al., 2024b) propose memory-efficient gradient approximations, and LoRA-One (Zhang et al., 2025b) employs the SVD of the full gradient for initialization.

Convergence guarantees for PEFT algorithms remain scarce, and existing results typically address only the low-dimensional parameters (Jiang et al., 2024). A related line of work studies *subspace descent* methods (Chen et al., 2025; He et al., 2024; Kozak et al., 2019; Liang et al., 2024), which constrain updates to $W_{k+1} \leftarrow W_k - \eta_k P_k P_k^\top G_k$, where $P_k$ is the projection matrix, $\eta_k$ the learning rate, and $G_k$ the full gradient. These approaches establish convergence in the full-parameter space, but under extra structural assumptions. For example, Liang et al. (2024) analyze a continuous-time variant via Lyapunov arguments, but require $P^\top G = 0 \Rightarrow G = 0$, which holds only if $P$ has full column rank—an unrealistic condition when $r < m$. Likewise, Chen et al. (2025); He et al. (2024); Kozak et al. (2019) rely on random projection theory, assuming $\mathbb{E}[PP^\top] = I_m$ and $P^\top P = I_r$, conditions not needed in our analysis. LDAdam (Robert et al., 2024) imposes a design assumption that the projection $P$ must produce a strict contraction. Closer to LoRA, Jang et al. (2024) provide a convergence analysis, but only within the Neural Tangent Kernel (NTK) regime, limiting its applicability.

Subspace minimization is a classical theme in nonlinear optimization (Conn et al., 1994; Cragg & Levy, 1969; Yuan, 2014). It was historically overshadowed by full-parameter algorithms such as L-BFGS (Liu & Nocedal, 1989) and conjugate gradient methods (Nocedal & Wright, 2006, Ch. 5), since many traditional applications could afford storing full gradients and quasi-Newton pairs. More recently, however, subspace-based strategies have re-emerged in large-scale derivative-free optimization (Cartis & Roberts, 2023; Dzahini & Wild, 2024; Menickelly, 2024; Nozawa et al., 2025; Zhang, 2025), where gradients are unavailable and low-dimensional surrogates are crucial. Furthermore, in the pre-training regime, several recent works have leveraged subspace techniques through projection-based methods, offering insights into effective subspace selection for LLM training. Ga-Lore (Zhao et al., 2024) constructs the projection subspace using a top-$r$ SVD of the full gradient, while Fira (Chen et al., 2024) augments this with a gradient-correction step to reduce projection bias. SARA (Zhang et al., 2025a) employs importance sampling to choose subspaces based on the SVD spectrum of the full gradient. APOLLO (Zhu et al., 2024) instead uses randomized projections to define the subspace, and SubTrack++ (Rajabi et al., 2025) explores Grassmannian subspaces for improved tracking.

## 2 PESO: A FRAMEWORK FROM SUBSPACE MINIMIZATION

In this section, we provide a novel perspective of PEFT methods with insights from subspace minimization. We summarize an algorithmic framework **P**arameter-**E**fficient **S**ubspace **O**ptimization (**PESO**) in Algorithm 1, and discuss how it unifies many benchmarks such as LoRA and GaLore.

To build an iterative scheme, a central question is how to represent the weight $W$ at each iteration using low-dimensional representation $\xi$. In (3), the optimization is expressed through evolving subspaces. At iteration $k$, we define the *anchored state* $\widetilde{W}_k := W_0 + \sum_{i=1}^{k-1} \mathcal{M}_i(\xi_i^*)$ to encode historical progress, and represent

$$W_k = \widetilde{W}_k + \mathcal{M}_k(\xi_k). \tag{4}$$

---

**Algorithm 1 PESO**: Generic Framework of **P**arameter-**E**fficient **S**ubspace **O**ptimization

---

**Require:** Initialization $W_0 \in \mathbb{R}^{m \times n}$, $\xi_0 \in \mathbb{R}^d$, and $\mathcal{M}_0$; an algorithmic subroutine UpdateSubspace, an optimizer Opt, frequency $K$.
1: Set $k \leftarrow 1$ and $\widetilde{W}_0 \leftarrow W_0$.
2: **while** stopping criteria not satisfied **do**
3:      $(\mathcal{M}_k, \widetilde{W}_k) \leftarrow (\mathcal{M}_{k-1}, \widetilde{W}_{k-1})$.
4:      **if** $k - 1 \bmod K = 0$ **then**            ▷ **Exploration** to new $\mathcal{S}_k$
5:          $(\mathcal{M}_k, \widetilde{W}_k) \leftarrow$ UpdateSubspace$(\mathcal{M}_{k-1}, \widetilde{W}_{k-1})$.
6:      **end if**
7:      $\Delta\xi_k \leftarrow$ Opt$(\xi_{k-1}, \mathcal{M}_k)$              ▷ **Exploitation** of current $\mathcal{S}_k$
8:      $\xi_k \leftarrow \xi_{k-1} + \Delta\xi_k$.
9:      $k \leftarrow k + 1$.
10: **end while**

---

Following the design of subspace minimization, PESO considers each $\mathcal{M}_k$ to admit a simple image in the form of a subspace: $\mathcal{S}_k := \{\mathcal{M}_k(\xi) : \xi \in \mathbb{R}^d\}$.

Under representation (4), the evolution of $W_k$ can be viewed as three complementary operations: 1) **exploration**: updating $\mathcal{M}_k$ to select a new subspace $\mathcal{S}_k$, (line 5 of Algorithm 1) 2) **exploitation**: optimizing $\xi_k$ within the current subspace (line 7-8 of Algorithm 1), and 3) updating $\widetilde{W}_k$ to absorb past progress into the anchored weights. These operations mirror the classical paradigm of **subspace minimization** (Conn et al., 1994), where a large-scale problem is solved by iteratively: (i) constructing a subspace based on local information such as gradients, (ii) solving a reduced subproblem within that subspace, and (iii) updating the iterate to incorporate the subspace solution.

In our design, exploration and exploitation directly parallel subspace selection and subproblem optimization, while the anchored state $\widetilde{W}_k$ retains progress from earlier subspaces. In LoRA, $\widetilde{W}$ is fixed at $W_0$, confining progress to the active subspace. In contrast, updating $\widetilde{W}$ absorbs accumulated contributions back into the parameter space, giving rise to two distinct exploration strategies: *warm-start* and *restart*, which we detail below in Section 2.1.

Leveraging this connection to subspace minimization, we present our generic framework PESO in Algorithm 1. It is important to note that, by selecting corresponding parameterization of $\mathcal{M}_k$, UpdateSubspace, and Opt, we are able to recover a variety of existing benchmarking methods in parameter-efficient training; see representatives in Table 1. We also remark that Algorithm 1 is equivalent to the classical two-loop subspace minimization scheme (Conn et al., 1994), which we defer to Appendix B in Algorithm 4.

## 2.1 SUBSPACE EXPLORATION-EXPLOITATION IN PESO

Now let us discuss two main components of our framework, subspace exploration and exploitation.

**Subspace Exploration.** Exploring new subspaces is essential for navigating the full-parameter space under memory restriction. Algorithm 1 carries out exploration by UpdateSubspace, which updates both $\mathcal{M}_k$ and $\widetilde{W}_k$. Such updates are often performed lazily every $K$ iterations, as in prior works (Liang et al., 2024; Zhang et al., 2023; Zhao et al., 2024; Zhu et al., 2024).

Depending on how much $\mathcal{M}_k$ is changed, two philosophies arise for how exploration interacts with the low-dimensional $\xi_k$: *warm-start* and *restart*. These are simply two modes of UpdateSubspace:

- **Warm-start.** Preserve $\xi_k$ and keep $\widetilde{W}_k$ fixed. Formally,

$$W_{k+1} = \widetilde{W}_k + \mathcal{M}_{k+1}(\xi_k + \Delta\xi_k). \tag{5}$$

- **Restart.** Absorb the previous contribution into the baseline, $\widetilde{W}_{k+1} \leftarrow \widetilde{W}_k + \mathcal{M}_k(\xi_k)$, and start the new subspace from $\xi_{\text{new}}$ (often 0):

$$W_{k+1} = \widetilde{W}_{k+1} + \mathcal{M}_{k+1}(\xi_{\text{new}} + \Delta\xi_k). \tag{6}$$

Intuitively, warm-start provides smoother transitions when consecutive subspaces remain similar, while restart marks a new phase, useful when the optimization geometry changes sharply. In prac-

Table 1: Examples of memory-efficient training methods as instances of PESO.

| Methods | $\xi$ | $\mathcal{M}_k(\xi)$ | $\mathcal{S}_k$ | `UpdateSubspace` | Init. |
|---|---|---|---|---|---|
| LoRA | $(A, B)$ | $AB$ | $\{A_k B + A B_k : A \in \mathbb{R}^{m \times r}, \ B \in \mathbb{R}^{r \times n}\}$ | Adam for $A_k, B_k$ | warm-start |
| AdaLoRA | $\Lambda$ | $P_k \Lambda Q_k$ | $\{P_k \Lambda Q_k : \Lambda \in \mathbb{R}^{r \times r} \text{ diagonal}\}$ | SGD for $P_k, Q_k$ | warm-start |
| GaLore | $R$ | $P_k R$ | $\{P_k R : R \in \mathbb{R}^{r \times n}\}$ | $P_k$: left $r$-SVD of $G_k$ | restart |
| Kozak et al. (2019) | $R$ | $P_k R$ | $\{P_k R : R \in \mathbb{R}^{r \times n}\}$ | randomly sample $P_k$ | restart |
| Liang et al. (2024) | $R$ | $P_k R$ | $\{P_k R : R \in \mathbb{R}^{r \times n}\}$ | online PCA of $P_k$ | warm-start |

tice, these two modes naturally lead to two main approaches for designing `UpdateSubspace`. A warm-start typically updates the parameterization of $\mathcal{M}_k$ smoothly along an *optimization trajectory*—for example, by applying an Adam step on subspace parameters as in LoRA variants—yielding a gradually evolving subspace. Restart, on the other hand, often *reassigns* $\mathcal{M}_k$ directly using local information such as gradients. This strategy is common in classical optimization; for example, in line search (a one-dimensional subspace method) each iteration resets the step size initialization when a new direction is chosen (Nocedal & Wright, 2006, Ch. 3). It is also used in LLM training, as in GaLore (Zhao et al., 2024), which periodically resets the subspace via the SVD of the full gradient. Concrete examples of both approaches are summarized in Table 1, and Section 3.1 introduces a new restart scheme leveraging full gradients.

**Subspace Exploitation.** Between two updates of `UpdateSubspace`, our framework performs $K$ iterations of `Opt` within the current subspace $\mathcal{S}_k$. This amounts to solving the subproblem

$$\min_{\xi \in \mathbb{R}^d} \ \ell(\widetilde{W}_k + \mathcal{M}_k(\xi)) \tag{7}$$

approximately for $K$ steps. In practice, `Opt` is often chosen as Adam.

The philosophy relies on a common belief in classical optimization: during an optimization procedure, once an effective subspace is identified, repeatedly exploiting it for multiple iterations improves efficiency. This principle underlies many classical optimization methods, such as trust-region methods (Nocedal & Wright, 2006, Ch. 4) and L-BFGS-B (Byrd et al., 1995).

## 2.2 Connection to Existing Benchmarks

While (4) may strike to be abstract, many existing benchmarks for LLM training can naturally fit in it by considering specific subspaces. Here we summarize several notable methods in Table 1.

• *Projected subspace.* A simple way to define a memory-efficient subspace is through low-rank projection, where $\mathcal{M}_k : R \in \mathbb{R}^{r \times n} \mapsto P_k R$ is parameterized by a left-projection matrix $P_k \in \mathbb{R}^{m \times r}$. This formulation can be extended to right-sided or two-sided projections. By applying the chain rule to $\nabla_R \ell(\widetilde{W}_k + P_k R)$, one obtains the projected subspace schemes analyzed in (He et al., 2024; Kozak et al., 2019; Liang et al., 2024; Zhao et al., 2024); see Appendix C for details. Within PESO, GaLore (Zhao et al., 2024), APOLLO (Zhu et al., 2024), and stochastic subspace descent (Kozak et al., 2019) correspond to a *restart* strategy by reassigning $P_k$, while online subspace descent (Liang et al., 2024) adopts a *warm-start* update of $P_k$ via online PCA.

• *Low-rank subspace.* The LoRA family defines the subspace $\{A_k B + A B_k : A \in \mathbb{R}^{m \times r}, B \in \mathbb{R}^{r \times n}\}$, where the adapters $(A, B)$ jointly serve as both $\xi$ and the parameterization of $\mathcal{M}_k$. Consequently, a single Adam update of $(A, B)$ simultaneously updates the subspace and its coordinates, effectively realizing a *warm-start* scheme with $K = 1$. Many LoRA variants can be viewed as modifications of this generic template: LoRA-Pro (Wang et al., 2024b) applies a different preconditioner, and PiSSA (Meng et al., 2024), LoRA-GA (Wang et al., 2024a), and LoRA-One (Zhang et al., 2025b) adjust initialization strategies, while other works modify learning rates or scaling rules. Our framework unifies these designs by interpreting them as *specific choices of* `Opt` *or initialization* within the same subspace structure.

• *SVD subspace.* A principled way to extract low-dimensional structure from matrices (such as $\Delta W$) is through Singular Value Decomposition (SVD), leading to the representation $\mathcal{M}_k : \lambda \in \mathbb{R}^r \mapsto$

$U\mathrm{diag}(\lambda)V$. Here, $(U, V)$ define the subspace (exploration), while $\lambda$ is the low-dimensional coordinates (exploitation). This separation fits directly into Algorithm 1, enabling flexible optimization strategies for $(U, V)$ and $\lambda$. AdaLoRA (Zhang et al., 2023) exemplifies this parameterization, and our framework clarifies the roles of $(P_k, \Lambda_k, Q_k)$ in their notation. We build on this principle in Section 3, where we propose a practical SVD-based variant `PESO-LoRA-T`.

Together, these three subspace categories illustrate how PESO unifies existing PEFT methods under a single framework, setting the stage for designing new practical algorithms and proving convergence in the following sections.

## 3 PESO-LoRA: Practical Algorithms from the Framework

The previous section establishes a conceptual link between PEFT methods and classical subspace minimization, providing a unifying interpretation. Building on this perspective, we now develop a concrete algorithm **PESO-LoRA**, which extends LoRA using guidance from our PESO framework. We present two variants: `PESO-LoRA-R` leverages a full gradient restart strategy to improve *exploration* of subspaces, and `PESO-LoRA-T` is a SVD-based method that enhances *exploitation* through more effective optimization within each subspace.

### 3.1 Full Gradient Restart

We now introduce an important variant of the `UpdateSubspace` subroutine in the restart category (see (6)) that enables convergence to stationarity in the full-parameter space. We design `UpdateSubspace` so that each new subspace $\mathcal{S}_k$ induced by $\mathcal{M}_k$ remains well aligned with the current full gradient $G_k$. We call this scheme *full gradient restart*:

**Full Gradient Restart.** Given learning rates $\{\eta_k\}$, whenever $k - 1 \bmod K = 0$:

1) Absorb history: $\widetilde{W}_k \leftarrow \widetilde{W}_{k-1} + \mathcal{M}_{k-1}(\xi_{k-1})$.

2) Compute the (stochastic) full gradient $G_k = \nabla_W \ell(\widetilde{W}_k)$.

3) Choose a low rank subspace $\mathcal{S}_k^{\mathrm{FG}}$ depending on $G_k$.

4) Restart with $\xi_k \leftarrow \xi_k^{\mathrm{new}}$ such that $\mathcal{M}_k(\xi_k^{\mathrm{new}}) = -\eta_k P_{\mathcal{S}_k^{\mathrm{FG}}}(G_k)$.

Here, $P_{\mathcal{S}_k^{\mathrm{FG}}}(G_k)$ denotes the projection of $G_k$ onto $\mathcal{S}_k^{\mathrm{FG}}$. This procedure effectively redefines $\mathcal{M}_k$ so that the new adapter is initialized by a *projected gradient step*:

$$W_k \leftarrow W_{k-1} - \eta_k P_{\mathcal{S}_k^{\mathrm{FG}}}(G_k). \tag{8}$$

Thus, each restart ensures that $\mathcal{S}_k$ captures information from full gradients, with initial progress comparable to a standard SGD step. In the literature on subspace methods, incorporating the full gradient into $\{\mathcal{S}_k\}$ is critical for convergence guarantees (Conn et al., 1994; Zhang, 2025). In particular, one can show that $\|\nabla_W \ell\| \to 0$ provided that $G_k := \nabla_W \ell(W_k)$ lies in $\mathcal{S}_k$. Building on this, we demonstrate in Section 5 that full gradient restart ensures convergence to a stationary point of the original problem (1) by interleaving projected steepest-descent steps with subspace updates.

---

**Algorithm 2** `PESO-LoRA-R`: **PESO** with **LoRA** and Subspace Explo**R**ation

---

**Require:** Pre-trained parameters $W_0 \in \mathbb{R}^{m \times n}$, frequency $K$, scale parameter $\gamma$.

1: Set $k \leftarrow 1$, $\widetilde{W}_0 \leftarrow W_0$, $A_0 \leftarrow 0$ and $B_0 \leftarrow 0$.
2: **while** stopping criteria not satisfied **do**
3:     **if** $k - 1 \bmod K = 0$ **then**
4:         $\widetilde{W}_k \leftarrow \widetilde{W}_{k-1} + A_{k-1}B_{k-1}$.
5:         Compute stochastic full gradient $G_k$.
6:         $(U_k, \Lambda_k, V_k) \leftarrow \mathrm{SVD}(-G_k)$.                      ▷ Top-$r$ SVD of $G_k$
7:         Set $A_{k-1} \leftarrow \frac{1}{\sqrt{\gamma}}U_k\sqrt{\Lambda_k}$ and $B_{k-1} \leftarrow \frac{1}{\sqrt{\gamma}}\sqrt{\Lambda_k}V_k$.
8:     **end if**
9:     $(A_k, B_k) \leftarrow \mathrm{AdamW}(A_{k-1}, B_{k-1})$.       ▷ One AdamW step on $(A_{k-1}, B_{k-1})$
10:    $k \leftarrow k + 1$.
11: **end while**
12: **return** $\widetilde{W}_k + A_k B_k$.

---

A practical construction of $\mathcal{S}_k^{\mathrm{FG}}$ is to compute a rank-$r$ SVD of $G_k$ and define the subspace as the span of its top singular directions. This ensures that $\mathcal{S}_k^{\mathrm{FG}}$ captures the main structure of $G_k$, while the approximation error $\|G_k - P_{\mathcal{S}_k^{\mathrm{FG}}}(G_k)\|$ is governed by the spectral tail of $G_k$. Crucially, this tail is *independent* of the rank gap in the objective, underscoring a key distinction between representation deficiency (e.g., LoRA) and update efficiency. In practice, given $(U_k, V_k)$ from the rank-$r$ SVD of $G_k$, one can also restart with $\mathcal{M}_k(\xi_k^{\mathrm{new}}) = -\eta_k U_k V_k$, which corresponds to the update from the recent training benchmark Muon (Jordan et al., 2024), providing improved stability over (8).

One practical advantage of full gradient restart is that it acts as a **"plug-and-play"** mechanism for existing PEFT methods. It can be applied with a moderate frequency $K$ to reduce the cost of SVD while still guiding subspace exploration effectively. Recent work, such as GaLore (Zhao et al., 2024) and LoRA-One (Zhang et al., 2025b), has demonstrated the empirical benefits of leveraging the full gradient. In particular, the recent variants LoRA-GA (Wang et al., 2024a) and LoRA-One (Zhang et al., 2025b) can be interpreted as special cases of LoRA with full gradient restart applied *only at initialization*. To achieve convergence in the full-parameter space, we propose PESO-LoRA-R (Algorithm 2), which embeds LoRA with a repeated restart mechanism every $K$ iterations. A detailed version of the pseudocode can be found in Appendix E.

To implement Algorithm 2, directly assigning $(A_{k-1}, B_{k-1}) \leftarrow 1/\sqrt{\gamma}(U_k\sqrt{\Lambda_k}, \sqrt{\Lambda_k}V_k)$ can cause instability due to mismatches in optimization states. A similar instability issue has also been reported for GaLore (Chen et al., 2024). For robustness, we propose alignment techniques to maintain consistency of subspace bases, momentum, and velocity. In particular, restarts often produce gradients with much larger magnitudes, leaving the Adam velocity "too cold" and causing unstable steps. We therefore apply a lightweight velocity alignment, rescaling the velocity to match the new gradient magnitude, $v \leftarrow \|g\|^2/\|v\| \, v$, together with a short $\beta_2$ warm-up, which we find to be the most important component for stable performance in practice; see more details in Appendix D. Finally, we remark both empirical evidence and theoretical results suggest that gradients $G_k$ in deep learning often have strong low-rank structure, making them especially suitable for efficient SVD-based approximations (Cosson et al., 2023; Yang et al., 2023; Zhao et al., 2024).

## 3.2 Exploitation via SVD Subspace

Having discussed exploration techniques inspired by subspace minimization, we now turn to the complementary philosophy: exploitation within the current subspace.

As outlined in Section 2.2, an SVD-based parameterization provides a clean and principled way to define $\mathcal{M}_k$. Specifically, we approximate the target mapping $\mathcal{M}(\xi^*)$ by a sum of rank-$r$ components, $\sum_k U_k \xi_k^* V_k$, where each pair $(U_k, V_k)$ defines an *SVD subspace* of rank $r$. Because SVD naturally captures the dominant gradient directions, this parameterization ensures that exploitation is focused on the most informative directions in the weight space.

Within each subspace, we optimize the low-dimensional $\xi$ for $K$ steps using AdamW. This design can be viewed as an extension of LoRA, with the key difference that the SVD structure explicitly decouples subspace exploitation (through $\xi$) from exploration (through $(U, V)$). The practical variant is summarized in Algorithm 3. A small frequency $K$ (e.g., 1 or 2) often suffices for strong performance without significant overhead. A detailed version of Algorithm 3 can be found in Appendix E.

---

**Algorithm 3** PESO-LoRA-T: **PESO** with **LoRA** and Subspace Exploi**T**ation

---

**Require:** Pretrained weights $W_0 \in \mathbb{R}^{m \times n}$, initial subspace matrices $U_0 \in \mathbb{R}^{m \times r}$, $V_0 \in \mathbb{R}^{r \times n}$, initial coordinate $\xi_0 \in \mathbb{R}^r$, frequency $K$.
1: Set $k \leftarrow 1$.
2: **while** stopping criterion not met **do**
3:      Keep $(U_k, V_k) \leftarrow (U_{k-1}, V_{k-1})$.
4:      **if** $k - 1 \mod K = 0$ **then**
5:          $(U_k, V_k) \leftarrow \mathtt{AdamW}(U_{k-1}, V_{k-1})$.             ▷ One AdamW step on $(U_{k-1}, V_{k-1})$
6:      **end if**
7:      $\xi_k \leftarrow \mathtt{AdamW}(\xi_{k-1})$.                            ▷ One AdamW step on $\xi_{k-1}$
8:      $k \leftarrow k + 1$.
9: **end while**
10: **return** $W_0 + U_k \operatorname{diag}(\xi_k) V_k$.

---

## 4 EXPERIMENTS

In this section, we conduct experiments to evaluate our methods across diverse tasks and models, comparing with standard LoRA-based approaches and full fine-tuning. We first assess natural language understanding on the GLUE benchmark (Wang et al., 2018) by fine-tuning T5-base (Raffel et al., 2020). We then evaluate natural language generation on Llama-2-7B (Touvron et al., 2023) for tasks including mathematical reasoning, code generation, and general instruction tuning. Finally, we demonstrate that `LoRA-PESO-R` remains effective even under strict memory constraints when trained for more epochs. Implementation details are provided in Appendix F.

### 4.1 NATURAL LANGUAGE UNDERSTANDING TASKS

We fine-tune the T5-base model on a subset of GLUE, including MNLI, SST-2, CoLA, QNLI, and MRPC, and evaluate performance using test accuracy (%). Following the setting in (Zhang et al., 2025b), we compare our method against several LoRA variants, including LoRA (Hu et al., 2022), LoRA+ (Hayou et al., 2024), P-LoRA (Zhang & Pilanci, 2024), PiSSA (Meng et al., 2024), LoRA-GA (Wang et al., 2024a), LoRA-Pro (Wang et al., 2024b), and LoRA-One (Zhang et al., 2025b). For fairness, hyperparameters are tuned individually for each method.

The results are summarized in Table 2. `PESO-LoRA-R` and `PESO-LoRA-T` achieve the best performance on three of the five GLUE tasks (MNLI, SST-2, and QNLI), which are also the *larger datasets*. On the remaining tasks, `PESO-LoRA-T` ranks second. This demonstrates the overall efficiency and robustness of our approaches, with advantages most evident on larger datasets that demand longer training and stronger exploration–exploitation. Moreover, `PESO-LoRA-T` generally outperforms `PESO-LoRA-R`, but at the cost of $1.4\times$ more computation time, whereas `PESO-LoRA-R` runs at nearly the same speed as standard LoRA. Memory costs are comparable across all methods, so the choice ultimately depends on whether performance or efficiency is prioritized.

Table 2: Performance of fine-tuned T5-base on natural language understanding tasks with rank set to 8. Results are reported as accuracy (%) over 3 runs. **Bold** and underline indicate the highest and second-highest accuracies *excluding* `PESO-LoRA-T`, which is shaded in gray and omitted from direct comparison due to its longer runtime.

| Method | MNLI | SST-2 | CoLA | QNLI | MRPC |
|---|---|---|---|---|---|
| LoRA | $85.30_{\pm0.04}$ | $94.04_{\pm0.09}$ | $72.84_{\pm1.25}$ | $93.02_{\pm0.07}$ | $68.38_{\pm0.01}$ |
| LoRA+ | $85.81_{\pm0.09}$ | $93.85_{\pm0.24}$ | $77.53_{\pm0.20}$ | $93.14_{\pm0.03}$ | $74.43_{\pm1.39}$ |
| P-LoRA | $85.28_{\pm0.15}$ | $93.88_{\pm0.11}$ | $79.58_{\pm0.67}$ | $93.00_{\pm0.07}$ | $83.91_{\pm1.16}$ |
| PiSSA | $85.75_{\pm0.07}$ | $94.07_{\pm0.06}$ | $74.27_{\pm0.39}$ | $93.15_{\pm0.14}$ | $76.31_{\pm0.51}$ |
| LoRA-GA | $85.70_{\pm0.09}$ | $94.11_{\pm0.18}$ | $80.57_{\pm0.20}$ | $93.18_{\pm0.06}$ | $85.29_{\pm0.24}$ |
| LoRA-Pro | $\underline{86.03}_{\pm0.19}$ | $94.19_{\pm0.13}$ | $\underline{81.94}_{\pm0.24}$ | $\underline{93.42}_{\pm0.05}$ | $\underline{86.60}_{\pm0.14}$ |
| LoRA-One | $85.89_{\pm0.08}$ | $\underline{94.53}_{\pm0.13}$ | $\mathbf{82.04}_{\pm0.22}$ | $93.37_{\pm0.02}$ | $\mathbf{87.83}_{\pm0.37}$ |
| `PESO-LoRA-R` | $\mathbf{86.08}_{\pm0.15}$ | $\mathbf{94.61}_{\pm0.09}$ | $81.50_{\pm0.16}$ | $\mathbf{93.43}_{\pm0.06}$ | $86.36_{\pm0.11}$ |
| `PESO-LoRA-T` | $86.09_{\pm0.04}$ | $94.76_{\pm0.19}$ | $82.01_{\pm0.30}$ | $93.45_{\pm0.03}$ | $87.59_{\pm0.46}$ |

### 4.2 NATURAL LANGUAGE GENERATION TASKS

Following prior work (Wang et al., 2024a; Zhang et al., 2025b), we fine-tune the Llama-2-7B model on three datasets and evaluate on the corresponding downstream tasks. For mathematical reasoning, we use a 100k subset of MetaMathQA (Yu et al., 2023) and evaluate on GSM8 (Cobbe et al., 2021). For general instruction tuning, we fine-tune on Alpaca (Taori et al., 2023) and evaluate on MMLU (Hendrycks et al., 2020). For code generation, we use a 100k subset of Code-Feedback (Zheng et al., 2024a) and evaluate on HumanEval (Chen et al., 2021), reporting PASS@1. To ensure fairness, all datasets are preprocessed to exclude overlaps with test sets. Results are shown in Table 3. Remarkably, our methods outperform baselines on two of the three tasks—mathematical reasoning and code generation—both involving *larger training datasets*, highlighting the substantial gains enabled by subspace exploration and exploitation in handling complex tasks.

For the more advanced model, we run the same three tasks by fine-tuning the Llama-3.1-8B (Dubey et al., 2024) model, comparing the LoRA-based methods with `PESO-LoRA-R`, and report the re-

Table 3: Performance of fine-tuned Llama-2-7B on natural language generation tasks with rank set to 8. Results are reported as accuracy (%) over 3 runs. **Bold** and underline indicate the highest and second-highest accuracies *excluding* PESO-LoRA-T.

|  | LoRA | LoRA-GA | LoRA-One | PESO-LoRA-R | PESO-LoRA-T |
|---|---|---|---|---|---|
| GSM8K | $59.26_{\pm 0.99}$ | $56.44_{\pm 1.15}$ | $60.44_{\pm 0.17}$ | **$60.55_{\pm 0.34}$** | $60.82_{\pm 0.77}$ |
| MMLU | $45.73_{\pm 0.30}$ | $45.15_{\pm 0.57}$ | **$47.24_{\pm 0.20}$** | $46.16_{\pm 0.58}$ | $46.44_{\pm 0.37}$ |
| HumanEval | $25.85_{\pm 1.75}$ | $26.95_{\pm 1.30}$ | $28.66_{\pm 0.39}$ | **$31.70_{\pm 1.30}$** | $30.85_{\pm 1.18}$ |

Table 4: Performance of fine-tuned Llama-3.1-8B on natural language generation tasks with rank set to 8. Results are reported as accuracy (%) over 3 runs. **Bold** and underline indicate the highest and second-highest accuracies.

|  | LoRA | LoRA-GA | LoRA-One | PESO-LoRA-R |
|---|---|---|---|---|
| GSM8K | $70.64_{\pm 0.53}$ | $76.67_{\pm 0.31}$ | $77.71_{\pm 0.17}$ | **$77.79_{\pm 0.18}$** |
| MMLU | $63.95_{\pm 0.05}$ | $62.91_{\pm 0.08}$ | $64.33_{\pm 0.14}$ | **$64.34_{\pm 0.21}$** |
| HumanEval | $42.47_{\pm 2.56}$ | $44.32_{\pm 5.64}$ | $45.32_{\pm 1.52}$ | **$47.15_{\pm 0.76}$** |

sults in Table 4. We find that on a more capable model, our method consistently outperforms all LoRA variants across all benchmarks.

Additionally, we compare our method with several pretraining-oriented approaches. Following the experimental setup in Zhu et al. (2024), we evaluate PESO-LoRA-R against GaLore (Zhao et al., 2024), Fira (Chen et al., 2024), and APOLLO (Zhu et al., 2024) by fine-tuning Llama-3-8B-Instruct (AI@Meta, 2024) on the Alpaca-en-demo dataset from LlamaFactory (Zheng et al., 2024b) with rank $r = 8$, and report the results on MMLU subtasks. As shown in Table 5, our PEFT-specialized method PESO-LoRA-R consistently outperforms these pretraining-oriented baselines in the fine-tuning regime.

We further conduct ablation studies by varying the rank $r$ and the restart frequency $K$. The results, reported in Appendix F.3, show that PESO-LoRA-R does not require a large rank to achieve strong performance and exhibits robust behavior across a wide range of restart frequencies. Memory and runtime comparisons are also provided in Appendix F.3, showing that PESO-LoRA-R has similar memory footprint as vanilla LoRA, and that their overall computational costs are comparable.

Table 5: Comparison with pretraining-oriented subspace methods. We fine-tune Llama-3-8B-Instruct on Alpaca-en-demo with rank $r = 8$ and evaluate on MMLU subtasks. Results are reported as accuracy (%). For GaLore, Fira, and APOLLO, we use the results reported in Zhu et al. (2024). Following their convention, we report our method's average accuracy, using the average over 3 runs. **Bold** and underline indicate the highest and second-highest accuracies.

| Method | STEM | Social Sciences | Humanities | Others | Average |
|---|---|---|---|---|---|
| GaLore | 54.50 | 75.11 | 58.59 | 72.03 | 64.43 |
| Fira | 53.53 | 75.46 | 58.59 | 72.09 | 64.32 |
| APOLLO /w SVD | 54.73 | 75.46 | 58.72 | 72.68 | 64.76 |
| APOLLO | 54.37 | 75.86 | 58.18 | 71.69 | 64.35 |
| PESO-LoRA-R | **57.15** | **76.86** | **60.79** | **73.57** | **66.47** |

## 4.3 MULTI-EPOCH LOW-RANK ANALYSIS

To demonstrate the effectiveness of subspace exploration, we extend T5-base fine-tuning on SST-2 from one epoch (Section 4.1) to four. This longer schedule enables more thorough exploration and reduces the intrinsic low-rank bottleneck. As shown in Table 6, PESO-LoRA-R with $r = 2$ consistently outperforms standard LoRA even at higher ranks ($r = 4, 8$), showing that it overcomes the low-rank limitation, achieves full-parameter optimality, and delivers stronger performance even under highly restricted memory budgets.

Table 6: Performance of fine-tuned T5-base (4 epochs) on the SST-2 dataset. Results are reported as accuracy (%) over 3 runs. **Bold** and underline indicate the highest and second-highest accuracies.

| Method | Epoch 1 | Epoch 2 | Epoch 3 | Epoch 4 |
|---|---|---|---|---|
| LoRA ($r = 2$) | $93.02_{\pm 0.44}$ | $93.47_{\pm 0.51}$ | $93.41_{\pm 0.12}$ | $93.52_{\pm 0.17}$ |
| LoRA ($r = 4$) | $94.23_{\pm 0.30}$ | $94.42_{\pm 0.15}$ | $94.46_{\pm 0.10}$ | $94.61_{\pm 0.25}$ |
| LoRA ($r = 8$) | $93.85_{\pm 0.30}$ | $94.03_{\pm 0.09}$ | $94.54_{\pm 0.05}$ | $94.54_{\pm 0.23}$ |
| `PESO-LoRA-R` ($r = 2$) | $\underline{94.30}_{\pm 0.15}$ | $\underline{94.47}_{\pm 0.08}$ | $\underline{94.84}_{\pm 0.25}$ | $\mathbf{95.14}_{\pm 0.15}$ |
| Full fine-tuning | $\mathbf{94.42}_{\pm 0.11}$ | $\mathbf{94.70}_{\pm 0.10}$ | $\mathbf{94.85}_{\pm 0.11}$ | $\underline{94.90}_{\pm 0.06}$ |

## 5 CONVERGENCE ANALYSIS

In this section, we establish convergence of PESO with the full gradient restart. For general nonconvex losses, the optimality measure $\mathbb{E}\|G_k\|$ converges to zero up to controlled inexactness. We begin by stating the regularity assumptions.

**Assumption 1.** The loss $\ell$ is nonconvex, bounded from below, and has $L$-Lipschitz gradients.

**Assumption 2.** Stochastic gradients $\widetilde{G}_k$ of the full gradient $G_k$ satisfy $\mathbb{E}(\widetilde{G}_k) = G_k$ and there exists $C > 0$ such that $\mathbb{V}(\widetilde{G}_k) \leq C$.

**Assumption 3.** The learning rates for full gradient restart in (8) satisfies $\sum \eta_k = \infty$ and $\sum \eta_k^2 < \infty$.

**Assumption 4.** There exists a sequence $\{\delta_k\} \geq 0$ such that $dist(G_k, \mathcal{S}_k) \leq \delta_k$ for $k$ where full gradient restart is implemmented. Furthermore, $\lim_{k \to \infty} \delta_k < \infty$.

**Assumption 5.** `Opt` and `UpdateSubspace` generate the updates satisfying $\mathbb{E}[\ell(W_k)] \leq \mathbb{E}[\ell(\widetilde{W}_k + \mathcal{M}_k(\xi_{k-1}))] + C_k$ and $\mathbb{E}[\ell(\widetilde{W}_k + \mathcal{M}_k(\xi_{k-1}))] \leq \mathbb{E}[\ell(W_{k-1})] + C_k$ where $\sum_k |C_k| < \infty$.

Assumptions 1–3 are standard in stochastic optimization; see, e.g., (Bottou et al., 2018). Assumption 4 requires the subspace at full gradient restart to be sufficiently aligned with $G_k$, allowing approximation errors, e.g., from low-rank SVD. Assumption 5 is mild and holds, for example, when both `Opt` and `UpdateSubspace` use SGD with diminishing learning rates; see Appendix G.

We now state the main convergence result; the proof and its deterministic counterpart are deferred to Appendix G. Importantly, the result holds for **any choice** of `UpdateSubspace`, whether warm-start or restart, and at **any frequency**. In particular, Algorithm 1 may combine different `UpdateSubspace` strategies at varying frequencies, and the guarantee remains valid.

**Theorem 5.1.** *Suppose all assumptions hold. With full gradient restart, the iterates $\{W_k\}$ generated by Algorithm 1 satisfy* $\liminf_{k \to \infty} \mathbb{E}[\|G_k\|] \leq \lim_{k \to \infty} \delta_k$.

*Remark* 1. If $\mathcal{S}_k^{\text{FG}}$ is chosen such that $G_k \in \mathcal{S}_k^{\text{FG}}$, then $\delta_k = 0$ and the optimality measure converges to zero. In `PESO-LoRA-R`, $\mathcal{S}_k^{\text{FG}}$ is the top-$r$ SVD subspace of $\widetilde{G}_k$, so $\delta_k$ reflects both SVD truncation error and a noise term of order $O(\sqrt{C})$. When $G_k$ is effectively low rank and variance-reduction (e.g., online PCA or EMA) is used, $\lim_{k \to \infty} \delta_k$ can be made small.

*Remark* 2. As a simple instantiation of PESO, we note that in the deterministic setting where $\mathcal{S}_k^{\text{FG}}$ is chosen as the top-$r$ SVD subspace of $G_k$, as in `PESO-LoRA-R`, the bias term in Theorem 5.1 disappears. In this case, we can establish exact asymptotic convergence, $\liminf_{k \to \infty} \mathbb{E}[\|G_k\|] = 0$, i.e., `PESO-LoRA-R` converges to a first-order stationary point; see full details in Appendix G.3.

## 6 CONCLUSIONS AND LIMITATIONS

This paper bridges classical methodology from nonlinear optimization with the practical challenge of memory-efficient LLM fine-tuning. It highlights two key perspectives: 1) practical constraints in LLM training, such as memory limits, can motivate specialized optimization designs; 2) principles from nonlinear optimization can in turn guide the development of practical algorithms for LLMs. We believe this opens promising directions for principled and scalable LLM training, while underscoring a broader philosophy: the rapid progress in LLMs can be enriched by classical foundations in computation and optimization. Our study has certain limitations. Due to limited resources, our experiments are restricted to medium-scale settings and do not yet reach the largest practical regimes. Extending our framework to full-scale pre-training remains an important future work, and we expect the methodology developed here to provide a solid foundation for such efforts.

ETHICS STATEMENT

Our research fully adheres to the ICLR Code of Ethics. We did not use any human or animal subjects. All datasets and models were acquired and used in accordance with their respective usage guidelines, and no private data was compromised. This work is free from bias and discriminatory outcomes, avoids using personally identifiable information, and presents no risks to privacy or security. We are committed to conducting this research with complete transparency and integrity.

REPRODUCIBILITY STATEMENT

We take reproducibility seriously and are willing to provide all necessary materials to support it. All theoretical results are presented with explicit assumptions, and full proofs are provided in Appendix G. Additionally, experimental settings and implementation details are documented in Appendix A, D and F. Together, these resources allow our claims and results to be verified and reproduced.

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

## A    SYNTHETIC EXAMPLE OF LoRA DEFICIENCY

One critical limitation in the literature is the absence of convergence guarantees toward valid optimality conditions of (1). Most existing works establish convergence only with respect to the low-dimensional parameters—such as the factors $A$ and $B$ in LoRA—but do not address convergence with respect to the full-parameter matrix $W$. For instance, Jiang et al. (2024) shows that $\nabla_A \ell(W_0 + A_k B_k)$ and $\nabla_B \ell(W_0 + A_k B_k)$ vanish as $k \to \infty$, while leaving the behavior of $\nabla_W \ell$ uncharacterized. This gap is not merely technical: it highlights a fundamental deficiency of PEFT methods compared to standard full-parameter training. In fact, the optimal loss attained by LoRA can be arbitrarily worse than the true optimal loss of (1). To illustrate this, consider the following simple synthetic example in matrix optimization:

$$\min_{W \in \mathbb{R}^{n \times n}} \quad \|W - M\|_F^2, \quad \text{where } M = a \cdot \mathrm{diag}(1, \ldots, 1, 0, \ldots, 0), \quad (r{+}1 \text{ ones}). \quad (9)$$

The optimal solution is clearly $W^* = M$ with $f(W^*) = 0$. However, applying LoRA with rank $r$ to (9) can at most achieve a rank-$r$ approximation of $M$, and attains $f(A^*B^*) = a^2$. As $a$ increases, or as the rank mismatch between the LoRA adapters and the true solution grows, the optimality gap between LoRA and full gradient methods can become arbitrarily large.

This example underscores the cost of memory restrictions: while low-rank parameterizations save memory, they may fundamentally limit convergence to the true optimum. In the middle and right panels of Figure 1, we show that LoRA with exploration (`PESO-LoRA-R`) can effectively converge to the true optimal solution while LoRA would not. Note in this implementation of `PESO-LoRA-R` (Algorithm 2), the SVD of the full gradient is a rank-$r$ SVD and therefore the low-rankness of this computational scheme would not affect the convergence.

## B    REVIEW ON SUBSPACE MINIMIZATION

It is worth noting that Algorithm 1 is essentially equivalent to the classical two-loop subspace minimization scheme (Conn et al., 1994), summarized in Algorithm 4.

The key distinction between warm-start and restart, discussed in (5) and (6) of Section 2.1, lies in how the subproblem (10) is initialized within each outer iteration of Algorithm 4.

## C    PROJECTED SUBSPACE AND MEMORY EFFICIENCY

One-sided projected subspaces in PESO can offer stronger memory efficiency than LoRA. This idea is exemplified by GaLore (Zhao et al., 2024), which we now place in the PESO framework. GaLore requires memory of order $mn + mr + 2nr$ (assuming $m \leq n$), compared to LoRA's $mn + 3mr + 3nr$.

---

**Algorithm 4** Classical Iterated-Subspace Minimization

---

**Require:** Initialization $W_0 \in \mathbb{R}^{m \times n}$, $\xi_0 \in \mathbb{R}^d$, $\mathcal{M}_0$, an algorithmic subroutine UpdateSubspace, an optimizer Opt, frequency $K$.

1: Set $k \leftarrow 1$ and $\widetilde{W}_0 \leftarrow W_0$.
2: **while** stopping criteria not satisfied **do**
3:     Update the subspace $(\mathcal{M}_k, \widetilde{W}_k) \leftarrow$ UpdateSubspace$(\mathcal{M}_{k-1}, \widetilde{W}_{k-1})$.
4:     Approximately solve the subspace minimization by Opt using $K$ inner-loop iterations:

$$\xi_k^* \leftarrow \text{approx} \arg\min_{\xi} \ \ell(\widetilde{W}_k + \mathcal{M}_k(\xi)) \tag{10}$$

5:     $W_k \leftarrow \widetilde{W}_k + \mathcal{M}_k(\xi_k^*)$.
6:     $k \leftarrow k + 1$.
7: **end while**

---

With the projected subspace representation in Table 1, optimization reduces to $\xi := R \in \mathbb{R}^{r \times n}$. By the chain rule,

$$\nabla_R \ell(W_k) = \nabla_\xi \ell(\widetilde{W}_k + P_k R_k) = P_k^\top \nabla_W \ell(W_k) = P_k^\top G_k. \tag{11}$$

Once $P_k$ is computed and stored, subspace gradients are obtained directly from $G_k$ with no extra overhead, though computing the full $G_k$ each iteration is more costly than subspace-only gradients.

Then, an *exploitation* step in the subspace by gradient descent with learning rate $\eta_k$ gives

$$\begin{aligned} W_{k+1} &= \widetilde{W}_k + \mathcal{M}_k(R_k - \eta_k P_k^\top G_k) \\ &= \widetilde{W}_k + P_k(R_k - \eta_k P_k^\top G_k) \\ &= W_k - \eta_k P_k P_k^\top G_k, \end{aligned} \tag{12}$$

which matches the classical projected subspace descent step (Kozak et al., 2019).

In GaLore, $P_k$ is chosen as the rank-$r$ left SVD of the full gradient at fixed intervals. PESO recovers GaLore when the subspace gradient $\nabla_R \ell(W_k) = P_k^\top G_k$ in (12) is replaced with its Adam update.

This shows how GaLore saves memory: gradients of $\xi_k$ are derived directly from $G_k$, and updates are written back into $W$ via (12), reusing the stored pretrained weights $W_0$. Thus explicit storage of $\xi_k$ is unnecessary. Further savings arise because GaLore omits optimizer states for $\mathcal{M}_k$ (i.e., for $P_k$), instead updating $P_k$ by direct reassignment in a restart manner. However, smoother transitions of subspace parameters often yield greater stability, as observed in our experiments; we discuss smoothing techniques for restart strategies in the next section.

# D    IMPLEMENTATION OF PESO-LoRA-R

We discuss several implementation details of PESO-LoRA-R that are critical for practical stability and performance.

## D.1    SMOOTHING THE SUBSPACES

A potential issue with restart methods (see (6)) is that they directly reassign the subspace parameterization from new information, which can introduce sharp changes and instability, especially in LLM training where stochastic noise is significant.

To mitigate this, we adopt an Exponential Moving Average (EMA) of old and new subspaces, similar in spirit to how Adam (Kingma & Ba, 2014) stabilizes noisy updates. However, this is nontrivial in PESO-LoRA-R (Algorithm 2), since the pre-restart adapters $(A_k, B_k)$—evolved through Adam dynamics—may differ significantly in scale and coordinates from the restarted pair $(U_k\sqrt{\Lambda_k}, \sqrt{\Lambda_k}V_k)$ obtained from rank-$r$ SVD. A naive EMA would mismatch these terms and discard valuable exploration information.

We resolve this by performing *basis and scaling alignment*. For clarity, we omit the subscript $k$. Given current adapters $(A, B)$, we first compute thin QR factorizations:

$$A = Q_A R_A, \quad B = R_B Q_B^\top, \quad Q_A^\top Q_A = I_r, \ Q_B^\top Q_B = I_r,$$

to extract bases $(Q_A, Q_B)$ and decouple scaling. Next, let the rank-$r$ SVD of the full gradient be $-G \approx U \Sigma V^\top$. We align $(U, V)$ to $(Q_A, Q_B)$ by applying SVD to $Q_A^\top U$ and $Q_B^\top V$:

$$Q_A^\top U = P_U \Sigma_U Q_U^\top, \quad R_U := P_U Q_U^\top, \qquad Q_B^\top V = P_V \Sigma_V Q_V^\top, \quad R_V := P_V Q_V^\top,$$

yielding aligned bases

$$\widehat{U} := U R_U^\top, \qquad \widehat{V} := V R_V^\top.$$

Here $R_U$ solves

$$\min_R \|UR - Q_A\|_F, \quad RR^\top = I,$$

so $\widehat{U}$ best aligns $U$ with $Q_A$ in Frobenius norm; the same holds for $\widehat{V}$. This produces the best alignment by classical low-rank SVD guarantees.

We then smooth the bases via EMA:

$$U_{\text{ema}} := \tau_1 Q_A + (1 - \tau_1)\widehat{U}, \qquad V_{\text{ema}} := \tau_1 Q_B + (1 - \tau_1)\widehat{V},$$

for smoothing parameter $\tau_1$. To smooth the scaling, we project the old adapter into the new bases and combine with the gradient:

$$S_{\text{new}} := \tau_2 \left[ U_{\text{ema}}^\top (AB) V_{\text{ema}} \right] - (1 - \tau_2) \left[ U_{\text{ema}}^\top G V_{\text{ema}} \right],$$

with parameter $\tau_2$. Since this merges the scaling of $A$ and $B$, we refactorize $S_{\text{new}}$ using polar decomposition (Glentis et al., 2025):

$$S_{\text{new}} = R_L \Sigma R_R^\top,$$

and define the new adapters via balanced splitting:

$$A_{\text{new}} := U_{\text{ema}} R_L \Sigma^{1/2}, \qquad B_{\text{new}} := \Sigma^{1/2} R_R^\top V_{\text{ema}}^\top.$$

Finally, although empirically less effective, we also rotate the momentum vectors according to the new basis. Specifically, we compute

$$T_A := Q_A^\top U_{\text{ema}}, \qquad T_B := Q_B^\top V_{\text{ema}},$$

and update the pre-restart momentum $(m_A, m_B)$ as

$$m_A \leftarrow m_A T_A, \qquad m_B \leftarrow T_B^\top m_B.$$

Training then proceeds with the new adapters $(A_{\text{new}}, B_{\text{new}})$ in `PESO-LoRA-R`.

## D.2 MOMENTUM AND VELOCITY ALIGNMENT

Even with basis and scaling alignment from the previous subsection, another stability issue arises: the new adapters $(A_{\text{new}}, B_{\text{new}})$ can induce gradients of very different magnitudes compared to the old $(A, B)$. Since restarts are based on the SVD of the full gradient, the new adapters align with top gradient directions, so the gradients with respect to $(A_{\text{new}}, B_{\text{new}})$ are typically larger. This mismatch can leave the velocity "too cold": historical states $(v_A, v_B)$ may underestimate the new gradient magnitudes, leading to an excessively large normalized step and unstable behavior, often observed as jumps in the loss curve.

To address this, we propose a combined *momentum/velocity scaling* technique with a $\beta_2$ *warm-up*. Let $(m_A, m_B)$ and $(v_A, v_B)$ denote the momentum and velocity before restart, and $(g_A, g_B)$ the gradients after restart computed with respect to $(A_{\text{new}}, B_{\text{new}})$. We define scaling factors

$$s_A^{(v)} = \frac{\|g_A\|^2}{\|v_A\|}, \quad s_A^{(m)} = \frac{\|g_A\|}{\|m_A\|}, \quad s_B^{(v)} = \frac{\|g_B\|^2}{\|v_B\|}, \quad s_B^{(m)} = \frac{\|g_B\|}{\|m_B\|},$$

which correct scale mismatches between the old optimization states and the new gradients. Here, $\|\cdot\|$ denotes the RMS norm. Momentum and velocity are then rescaled as

$$v_A \leftarrow s_A^{(v)} v_A, \quad m_A \leftarrow s_A^{(m)} m_A, \quad v_B \leftarrow s_B^{(v)} v_B, \quad m_B \leftarrow s_B^{(m)} m_B.$$

This resolves scale mismatches, but an additional adjustment is needed: $\beta_2 = 0.999$ (velocity EMA) adapts much more slowly than $\beta_1 = 0.9$ (momentum EMA). At initialization, bias correction balances these, but after a restart we require extra correction. We therefore decrease $\beta_2$ immediately after a restart and gradually warm it back to $0.999$ over a window $T$. If a restart occurs at iteration $t_r$, then for $t_r \leq t \leq t_r + T$ we set

$$\beta_2(t) = \beta_{2,\min} + \left( \beta_{2,\text{final}} - \beta_{2,\min} \right) \tfrac{1}{2} \left( 1 - \cos \tfrac{\pi(t - t_r)}{T} \right), \quad \beta_{2,\text{final}} = 0.999,$$

and for $t > t_r + T$ we use $\beta_2 = 0.999$ as usual. In our experiments, we set $\beta_{2,\min} = 0.95$ and $T = \lfloor K/3 \rfloor$.

---

**Algorithm 5** `PESO-LoRA-R`: **PESO** with **LoRA** and Subspace Explo**R**ation

---

**Require:** Pre-trained parameters $W_0 \in \mathbb{R}^{m \times n}$, frequency $K$, scale parameter $\gamma$, learning rate $\eta$, AdamW hyperparameters $(\beta_1, \beta_2, \varepsilon, \lambda)$.

1: Set $k \leftarrow 1$, $\widetilde{W}_0 \leftarrow W_0$, $A_0 \leftarrow 0$, $B_0 \leftarrow 0$.
2: Initialize AdamW states $m_A \leftarrow 0$, $v_A \leftarrow 0$, $m_B \leftarrow 0$, $v_B \leftarrow 0$.
3: **while** stopping criteria not satisfied **do**
4:     **if** $k - 1 \mod K = 0$ **then**
5:         $\widetilde{W}_k \leftarrow \widetilde{W}_{k-1} + A_{k-1}B_{k-1}$.
6:         Compute stochastic full gradient $G_k$.
7:         $(U_k, \Lambda_k, V_k) \leftarrow \text{SVD}(-G_k)$.                    $\triangleright$ Top-$r$ SVD of $G_k$
8:         Set $(A_{k-1}, B_{k-1}) \leftarrow (A_\text{new}, B_\text{new})$ according to Appendix D.1.
9:         Update $m_A, m_B, v_A, v_B, \beta_2$ according to Appendix D.2.
10:     **end if**
11:     Compute gradients $g_{A,k} \leftarrow \nabla_A \ell(\widetilde{W}_k + A_{k-1}B_{k-1})$, $g_{B,k} \leftarrow \nabla_B \ell(\widetilde{W}_k + A_{k-1}B_{k-1})$.
12:     $m_A \leftarrow \beta_1 m_A + (1 - \beta_1)g_{A,k}$,   $v_A \leftarrow \beta_2 v_A + (1 - \beta_2)g_{A,k} \odot g_{A,k}$.
13:     $m_B \leftarrow \beta_1 m_B + (1 - \beta_1)g_{B,k}$,   $v_B \leftarrow \beta_2 v_B + (1 - \beta_2)g_{B,k} \odot g_{B,k}$.
14:     $\widehat{m}_A \leftarrow m_A/(1 - \beta_1^k)$,   $\widehat{v}_A \leftarrow v_A/(1 - \beta_2^k)$.
15:     $\widehat{m}_B \leftarrow m_B/(1 - \beta_1^k)$,   $\widehat{v}_B \leftarrow v_B/(1 - \beta_2^k)$.
16:     $A_k \leftarrow A_{k-1} - \eta\lambda A_{k-1} - \eta\,\widehat{m}_A/(\sqrt{\widehat{v}_A} + \varepsilon)$.
17:     $B_k \leftarrow B_{k-1} - \eta\lambda B_{k-1} - \eta\,\widehat{m}_B/(\sqrt{\widehat{v}_B} + \varepsilon)$.
18:     $k \leftarrow k + 1$.
19: **end while**
20: **return** $\widetilde{W}_k + A_k B_k$.

---

**Algorithm 6** `PESO-LoRA-T`: **PESO** with **LoRA** and Subspace Exploi**T**ation

---

**Require:** Pretrained weights $W_0 \in \mathbb{R}^{m \times n}$, initial subspace matrices $U_0 \in \mathbb{R}^{m \times r}$, $V_0 \in \mathbb{R}^{r \times n}$, initial coordinate $\xi_0 \in \mathbb{R}^r$, frequency $K$, learning rate $\eta$, AdamW hyperparameters $(\beta_1, \beta_2, \varepsilon, \lambda)$.

1: Set $k \leftarrow 1$.
2: Initialize AdamW states $m_U, m_V, m_\xi \leftarrow 0$ and $v_U, v_V, v_\xi \leftarrow 0$.
3: **while** stopping criterion not met **do**
4:     Keep $(U_k, V_k) \leftarrow (U_{k-1}, V_{k-1})$.
5:     **if** $k - 1 \mod K = 0$ **then**
6:         Compute gradient $g_{U,k} \leftarrow \nabla_U \ell(W_0 + U_{k-1}\,\text{diag}(\xi_{k-1})V_{k-1})$.
7:         Compute gradient $g_{V,k} \leftarrow \nabla_V \ell(W_0 + U_{k-1}\,\text{diag}(\xi_{k-1})V_{k-1})$.
8:         $m_U \leftarrow \beta_1 m_U + (1 - \beta_1)g_{U,k}$,   $v_U \leftarrow \beta_2 v_U + (1 - \beta_2)g_{U,k} \odot g_{U,k}$.
9:         $m_V \leftarrow \beta_1 m_V + (1 - \beta_1)g_{V,k}$,   $v_V \leftarrow \beta_2 v_V + (1 - \beta_2)g_{V,k} \odot g_{V,k}$.
10:         $\widehat{m}_U \leftarrow m_U/(1 - \beta_1^k)$,   $\widehat{v}_U \leftarrow v_U/(1 - \beta_2^k)$.
11:         $\widehat{m}_V \leftarrow m_V/(1 - \beta_1^k)$,   $\widehat{v}_V \leftarrow v_V/(1 - \beta_2^k)$.
12:         $U_k \leftarrow U_{k-1} - \eta\lambda U_{k-1} - \eta\,\widehat{m}_U/(\sqrt{\widehat{v}_U} + \varepsilon)$.
13:         $V_k \leftarrow V_{k-1} - \eta\lambda V_{k-1} - \eta\,\widehat{m}_V/(\sqrt{\widehat{v}_V} + \varepsilon)$.
14:     **end if**
15:     Compute gradient $g_{\xi,k} \leftarrow \nabla_\xi \ell(W_0 + U_k\,\text{diag}(\xi_{k-1})V_k)$.
16:     $m_\xi \leftarrow \beta_1 m_\xi + (1 - \beta_1)g_{\xi,k}$,   $v_\xi \leftarrow \beta_2 v_\xi + (1 - \beta_2)g_{\xi,k} \odot g_{\xi,k}$.
17:     $\widehat{m}_\xi \leftarrow m_\xi/(1 - \beta_1^k)$,   $\widehat{v}_\xi \leftarrow v_\xi/(1 - \beta_2^k)$.
18:     $\xi_k \leftarrow \xi_{k-1} - \eta\lambda\xi_{k-1} - \eta\,\widehat{m}_\xi/(\sqrt{\widehat{v}_\xi} + \varepsilon)$.
19:     $k \leftarrow k + 1$.
20: **end while**
21: **return** $W_0 + U_k\,\text{diag}(\xi_k)V_k$.

---

# E   DETAILED VERSION OF ALGORITHMS

In this section, we provide detailed versions of `PESO-LoRA-R` (Algorithm 2) and `PESO-LoRA-T` (Algorithm 3).

## F  EXPERIMENTAL DETAILS

All experiments are conducted on NVIDIA RTX A6000 GPUs. For `PESO-LoRA-R`, to further reduce computational cost, we restrict the exploration frequency to at most two times per epoch.

### F.1  NATURAL LANGUAGE UNDERSTANDING: HYPERPARAMETER SETTINGS

In Section 4.1, we present the results of our methods and various LoRA-based algorithms on natural language understanding tasks, following the prompt tuning configuration of (Wang et al., 2024a). The general hyperparameter settings are kept consistent across all algorithms which are shown in Table 7. To ensure a fair comparison, we follow (Zhang et al., 2025b) and tune the learning rates via grid search over $\{1 \times 10^{-4}, 2 \times 10^{-4}, 5 \times 10^{-4}, 1 \times 10^{-3}\}$. Additionally, following the choices of Zhang et al. (2025b), the scale parameters for LoRA-One are set to be $\{128, 16, 128, 128, 64\}$ for {MNLI, SST-2, CoLA, QNLI, MRPC}.

For `PESO-LoRA-R`, we set the smoothing parameter $\tau_1 = \tau_2 = 0.9$, with frequency $K$ chosen as $\{2000, 500, 100, 500, 40\}$ for {MNLI, SST-2, CoLA, QNLI, and MRPC} based on empirical observations. When $(k - 1) \mod K = 0$ and $k \neq 0$, we set the scale parameter $\gamma = 1$; when $k = 0$, the scale parameter is set the same as in LoRA-One. To further reduce computational cost, we restrict the restart frequency to times per epoch. For `PESO-LoRA-T`, we set frequency $K = 1$ for all datasets.

Table 7: Common hyperparameters for LoRA fine-tuning on T5-base model.

| Epoch | Optimizer | $(\beta_1, \beta_2)$ | $\epsilon$ | Batch Size | Weight Decay | LR Scheduler |
|---|---|---|---|---|---|---|
| 1 | AdamW | (0.9, 0.999) | $1 \times 10^{-8}$ | 32 | 0 | cosine |
| Warm-up Ratio | LoRA Alpha | #Runs | Sequence Length | Adapt Precision | Backbone Precision | Gradient Batch Size |
| 0.03 | 16 | 3 | 128 | FP32 | FP32 | 8 |

### F.2  NATURAL LANGUAGE GENERATION: HYPERPARAMETER SETTINGS

#### F.2.1  COMPARISON WITH LoRA-BASED METHODS

For natural language generation tasks in Section 4.2, we follow the configuration of prompt tuning and strategy of hyperparameter tuning in (Zhang et al., 2025b) to ensure fair comparison. We search the best learning rate over $\{5 \times 10^{-4}, 2 \times 10^{-4}, 1 \times 10^{-4}, 5 \times 10^{-5}, 2 \times 10^{-5}, 1 \times 10^{-5}\}$, and the general hyperparameter setting is summarized in Table 8. Additionally, following the choice of Zhang et al. (2025b), the scale parameters are tuned within $\{16, 32, 64, 128\}$ for LoRA-GA, LoRA-One and our methods to achieve the best performances.

Table 8: Common hyperparameters for LoRA fine-tuning on Llama-2-7B and LLama-3.1-8B model.

| Epoch | Optimizer | $(\beta_1, \beta_2)$ | $\epsilon$ | Batch Size | Weight Decay | LR Scheduler |
|---|---|---|---|---|---|---|
| 1 | AdamW | (0.9, 0.999) | $1 \times 10^{-8}$ | 32 | 0 | cosine |
| Warm-up Ratio | LoRA Alpha | #Runs | Sequence Length | Adapter Precision | Backbone Precision | Gradient Batch Size |
| 0.03 | 16 | 3 | 1024 | FP32 | BF16 | 8 |

For `PESO-LoRA-R`, we set the smoothing parameter $\tau_1 = \tau_2 = 0.9$, with frequency $K = 500$ for all the experiments. When $(k - 1) \mod K = 0$ and $k \neq 0$, we set the scale parameter $\gamma = 1$; when $k = 0$, the scale parameter is set the same as in LoRA-One. For `PESO-LoRA-T`, we set frequency $K = 1$ for all datasets.

#### F.2.2  COMPARISON WITH PRETRAINING-ORIENTED SUBSPACE METHODS

For comparison with pretraining-oriented subspace-based approaches, we follow the experimental setup described in Zhu et al. (2024). Specifically, we fine-tune Llama-3-8B-Instruct on the Alpaca-en-demo dataset for 3 epochs and evaluate the resulting models on the MMLU subtasks. For GaLore (Zhao et al., 2024), Fira (Chen et al., 2024), APOLLO (Zhu et al., 2024), and our `PESO-LoRA-R`, we report the best accuracy achieved by sweeping the learning rate over

Table 9: Ablation on rank $r$ for `PESO-LoRA-R`, evaluated on GSM8K with restart frequency $K = 500$.

| Rank $r$ | 1 | 2 | 4 | 8 | 16 | 32 | 64 |
|---|---|---|---|---|---|---|---|
| Acc (%) | 75.28 | 75.97 | 76.88 | 77.56 | 77.71 | 78.01 | 78.39 |

Table 10: Ablation on restart frequency $K$ and restart count (once or twice per epoch), evaluated on GSM8K with rank $r = 8$. Infinity denotes applying exploration only at initialization.

| $K$ | 250 | 500 | 750 | 1000 | Infinity |
|---|---|---|---|---|---|
| Restart Once | 77.18 | 77.56 | 77.18 | 77.26 | 77.26 |
| Restart Twice | 76.88 | 77.30 | 77.56 | 76.88 | – |

$\{5e{-}6, 7.5e{-}6, 1e{-}5, 2.5e{-}5, 5e{-}5, 7.5e{-}5, 1e{-}4, 1.5e{-}4, 2e{-}4\}$. All methods use a fixed rank of $r = 8$. For `PESO-LoRA-R`, the scale parameter is set to $\gamma = 16$.

### F.3 NATURAL LANGUAGE GENERATION: ABLATION STUDY AND COMPUTATIONAL COST

We conduct an ablation study on `PESO-LoRA-R` by fine-tuning Llama-3.1-8B and evaluating the model on GSM8K. We vary the rank $r$ and the restart frequency $K$, and additionally compare using one versus two restarts per epoch. We set the scale parameter to $\gamma = 128$, and all other hyperparameters, unless otherwise noted, follow Appendix F.2. The results are presented in Table 9 and 10.

Overall, the results indicate that `PESO-LoRA-R` does not require a large rank to achieve strong performance: accuracy improves substantially even at small ranks, with diminishing gains beyond $r = 8$. Furthermore, the method shows robust behavior across a wide range of restart frequencies, as performance remains stable when varying $K$ or the number of restarts performed per epoch.

We also report memory and runtime for Llama-3.1-8B fine-tuning on GSM8K with rank $r = 8$, $K = 500$, and restricting one restart per epoch. As shown in Table 11, `PESO-LoRA-R` matches the memory footprint of vanilla LoRA and incurs only a negligible increase in runtime. Specifically, the SVD (incurred when doing restart) step accounts for only 1.9% of the total computation time in our experiments, indicating that its overhead is negligible.

### F.4 MULTI-EPOCH LOW-RANK ANALYSIS: HYPERPARAMETER SETTINGS

In Section 4.3, we fine-tune the T5-base model on SST-2 dataset for 4 epochs. We vary the rank of LoRA in $\{2, 4, 8\}$, keep the rank of `PESO-LoRA-R` as 2, and add full-parameter fine-tuning for comparison. We keep all the other hyperparameter settings the same as in F.1.

## G PROOFS

In this section, we provide the proofs of the theoretical results stated in Section 5. For completeness, we begin with the deterministic case, i.e., when the gradients $G_k$ accessed by PESO (Algorithm 1) are exact, without stochastic noise. We then prove Theorem 5.1, which considers the stochastic setting where only noisy gradients $\widetilde{G}_k$ are available, satisfying Assumption 2.

### G.1 DETERMINISTIC CASE

We first state the deterministic counterpart of Assumption 5:

**Assumption 6.** `Opt` and `UpdateSubspace` generate the updates satisfying $\ell(W_k) \leq \ell(\widetilde{W}_k + \mathcal{M}_k(\xi_{k-1}))$ and $\ell(\widetilde{W}_k + \mathcal{M}_k(\xi_{k-1})) \leq \ell(W_{k-1})$ for all $k = 1, 2, \cdots$.

Assumption 6 requires that both `Opt` and `UpdateSubspace` act as descent methods, ensuring that the loss is non-increasing. This is the deterministic analogue of Assumption 5, and it holds, for example, when both are implemented as gradient descent with step size $\alpha \leq 1/L$, where $L$ is the Lipschitz constant from Assumption 1. In particular, one can take `UpdateSubspace` to be a warm-start step that performs gradient descent on the subspace parameters with respect to the

Table 11: Memory (excluding transient peaks) and time for Llama-3.1-8B fine-tuning on GSM8K with $r = 8$, $K = 500$, and one restart per epoch.

| Method | LoRA | PESO-LoRA-R |
|--------|------|-------------|
| Time | 5h03m | 5h09m |
| Memory | 25.7G | 25.7G |

original loss function. This is formalized below in a standard result from the optimization literature (see, e.g., Nesterov, 2018).

**Proposition G.1.** *Let* `Opt` *and* `UpdateSubspace` *be gradient descent schemes on $\ell$ with constant learning rate $\alpha \leq 1/L$. Then Assumption 6 holds.*

We now present the deterministic convergence result.

**Theorem G.2.** *Suppose Assumptions 1, 4, and 6 hold. With full gradient restart enabled and learning rate $\eta_k = \frac{1}{L}$, the iterates $\{W_k\}$ generated by Algorithm 1 satisfy $\liminf_{k \to \infty} \|G_k\| \leq \lim_{k \to \infty} \delta_k$.*

*Proof.* We assume the frequency of the full gradient restart is $K$. By the descent lemma, whenever $k - 1 \mod K = 0$ (i.e., when a full gradient restart occurs), define

$$\widehat{W}_k := W_{k-1} - \frac{1}{L} P_{\mathcal{S}_k}(G_k). \tag{13}$$

Performing a full gradient restart as `UpdateSubspace`, as discussed in (8), is thus equivalent to moving from $W_{k-1}$ to $\widehat{W}_k$. It follows that

$$\begin{aligned} \ell(\widehat{W}_k) = \ell(W_{k-1} - \tfrac{1}{L} P_{\mathcal{S}_k}(G_k)) &\leq \ell(W_{k-1}) + \left\langle G_k, -\tfrac{1}{L} P_{\mathcal{S}_k}(G_k) \right\rangle + \tfrac{L}{2} \|\tfrac{1}{L} P_{\mathcal{S}_k}(G_k)\|^2 \\ &= \ell(W_{k-1}) - \tfrac{1}{L} \|P_{\mathcal{S}_k}(G_k)\|^2 + \tfrac{L}{2} \|\tfrac{1}{L} P_{\mathcal{S}_k}(G_k)\|^2 \\ &= \ell(W_{k-1}) - \tfrac{1}{2L} \|P_{\mathcal{S}_k}(G_k)\|^2, \end{aligned} \tag{14}$$

where the second equality holds because projection onto a subspace is orthogonal.

Therefore, by Assumption 6, for iterates $i = k, \ldots, k + K - 1$ (note that $\ell(W_k) \leq \ell(\widehat{W}_k)$ since $W_k$ is obtained by applying `Opt` to $\widehat{W}_k$),

$$\tfrac{1}{2L} \|P_{\mathcal{S}_k}(G_k)\|^2 \leq \ell(W_{k-1}) - \ell(\widehat{W}_k) \leq \ell(W_{k-1}) - \ell(W_i). \tag{15}$$

Importantly, (15) remains valid regardless of how frequently other types of `UpdateSubspace` are applied between full gradient restarts, since all updates preserve the descent property by Assumption 6. In particular, whenever $\mathcal{M}_k$ is updated without a full gradient restart, we have

$$\begin{aligned} \ell(W_{k-1}) - \ell(W_k) &= \ell(\widetilde{W}_{k-1} + \mathcal{M}_{k-1}(\xi_{k-1})) - \ell(W_k) \\ &\geq \ell(\widetilde{W}_{k-1} + \mathcal{M}_{k-1}(\xi_{k-1})) - \ell(\widetilde{W}_k + \mathcal{M}_k(\xi_{k-1})) \\ &\geq 0. \end{aligned} \tag{16}$$

This ensures the chain of inequalities in (15) continues to hold when updates are performed by `OptM`.

Hence, for any integer $k \in \mathbb{N}$,

$$\tfrac{1}{2L} \|P_{\mathcal{S}_{kK+1}}(G_{kK+1})\|^2 \leq \ell(W_{kK}) - \ell(W_{(k+1)K}). \tag{17}$$

Here $kK$ and $(k+1)K$ denote integer products.

Since $\{\ell(W_k)\}$ is bounded below (Assumption 1) and monotonically decreasing (Assumption 6 together with the descent lemma at restart points), it converges by the monotone convergence theorem and is Cauchy. Thus $\ell(W_k) - \ell(W_{k+1}) \to 0$, and

$$\tfrac{1}{2L} \|P_{\mathcal{S}_{kK+1}}(G_{kK+1})\|^2 \to 0. \tag{18}$$

Finally, note that

$$\|G_{kK+1}\| \leq \text{dist}(G_{kK+1}, \mathcal{S}_{kK+1}) + \|P_{\mathcal{S}_{kK+1}}(G_{kK+1})\| \leq \delta_{kK+1} + \|P_{\mathcal{S}_{kK+1}}(G_{kK+1})\| \to \delta, \tag{19}$$

where $\delta := \lim_{k \to \infty} \delta_k$. Therefore, $\liminf_{k \to \infty} \|G_k\| \leq \delta$. $\qquad\square$

## G.2 STOCHASTIC CASE

We begin by verifying the validity of Assumption 5. As an illustrative case, suppose `Opt` and `UpdateSubspace` are implemented by SGD with diminishing step sizes $\{\alpha_k\}$ satisfying $\sum_k \alpha_k < \infty$. Let $\widehat{W}_k$ denote the weight after such an update. By the descent lemma (see also (Bottou et al., 2018, Lemma 4.4)), the expected decrease can be bounded as

$$\mathbb{E}[\ell(\widehat{W}_k)] \leq \mathbb{E}[\ell(W_k)] - \alpha_k \left(1 - \frac{L\alpha_k}{2}\right)\mathbb{E}\|G_k\|^2 + \frac{L}{2}\alpha_k^2 C. \tag{20}$$

For $\alpha_k \leq 1/L$, this simplifies to

$$\mathbb{E}[\ell(\widehat{W}_k)] - \mathbb{E}[\ell(W_k)] \leq -\frac{\alpha_k}{2}\mathbb{E}\|G_k\|^2 + \frac{L}{2}\alpha_k^2 C. \tag{21}$$

Taking positive and negative parts, we obtain

$$\left[\mathbb{E}[\ell(\widehat{W}_k)] - \mathbb{E}[\ell(W_k)]\right]_+ \leq \frac{L}{2}C\alpha_k^2, \qquad \left[\mathbb{E}[\ell(W_k)] - \mathbb{E}[\ell(\widehat{W}_k)]\right]_+ \leq \mathbb{E}[\ell(W_k)] - \mathbb{E}[\ell(\widehat{W}_k)]. \tag{22}$$

Summing over all $k$,

$$\sum_{k=0}^{\infty} \left[\mathbb{E}[\ell(\widehat{W}_k)] - \mathbb{E}[\ell(W_k)]\right]_+ \leq \frac{L}{2}C\sum_{k=0}^{\infty}\alpha_k^2 < \infty,$$

$$\sum_{k=0}^{\infty} \left[\mathbb{E}[\ell(W_k)] - \mathbb{E}[\ell(\widehat{W}_k)]\right]_+ \leq \sup_k \mathbb{E}[\ell(W_k)] - \ell(W^*) < \infty. \tag{23}$$

Defining $C_k := \mathbb{E}[\ell(\widehat{W}_k)] - \mathbb{E}[\ell(W_k)]$, we conclude that $\sum_k |C_k| < \infty$, hence Assumption 5 holds.

We are now ready to present the proof of our main result, Theorem 5.1.

*Proof of Theorem 5.1.* Because $\mathcal{S}_k$ are subspaces, $P_{\mathcal{S}_k}$ is a linear operator, which allows the exchangability with $\mathbb{E}$. Therefore, one has

$$\begin{aligned}
\mathbb{E}(\|P_{\mathcal{S}_k}(\widetilde{G}_k)\|^2) &= \|\mathbb{E}(P_{\mathcal{S}_k}(\widetilde{G}_k))\|^2 + \mathbb{V}(P_{\mathcal{S}_k}(\widetilde{G}_k)) \\
&= \|(P_{\mathcal{S}_k}(G_k))\|^2 + \mathbb{E}\left(\|P_{\mathcal{S}_k}(\widetilde{G}_k - G_k)\|^2\right) \\
&\leq \|(P_{\mathcal{S}_k}(G_k))\|^2 + \mathbb{E}\left(\|\widetilde{G}_k - G_k\|^2\right) \\
&\leq \|(P_{\mathcal{S}_k}(G_k))\|^2 + C.
\end{aligned} \tag{24}$$

Similar to the deterministic case, We assume the frequency of the full gradient restart is $K$, and consider $k - 1 \mod K = 0$ (i.e., when a full gradient restart occurs). Again, we define

$$\widehat{W}_k := W_{k-1} - \frac{1}{L}P_{\mathcal{S}_k}(G_k). \tag{25}$$

By the property of the full gradient restart, we have

$$\begin{aligned}
\mathbb{E}[\ell(\widehat{W}_k)] &\leq \mathbb{E}[\ell(W_{k-1})] + \left\langle G_k, \eta_k \mathbb{E}[P_{\mathcal{S}_k}(\widetilde{G}_k)]\right\rangle + \frac{L}{2}\mathbb{E}[\|\eta_k P_{\mathcal{S}_k}(\widetilde{G}_k)\|^2] \\
&\leq \mathbb{E}[\ell(W_{k-1})] + \eta_k\|P_{\mathcal{S}_k}(G_k)\|^2 + \frac{L\eta_k^2}{2}(\|(P_{\mathcal{S}_k}(G_k))\|^2 + C) \\
&= \mathbb{E}[\ell(W_{k-1})] - \left(\eta_k - \frac{L\eta_k^2}{2}\right)\|P_{\mathcal{S}_k}(G_k)\|^2 + \frac{L\eta_k^2}{2}C,
\end{aligned} \tag{26}$$

where the first inequality follows from Assumption 1, and the second follows from the fact that $\mathcal{S}_k$ is a subspace and (24).

Then by Assumption 5, suppose $k - 1 \mod K = 0$, and for iterates $i = k+1, \cdots, k+K-1$,

$$\mathbb{E}[\ell(W_i)] \leq \mathbb{E}[\ell(W_{i-1})] + 2C_i, \tag{27}$$

where $2C_i$ comes from bounding the scenario where both `Opt` and `UpdateSubspace` operate at $i$-th iterate. Then summing up the inequalities (26) and (27) for $i = k+1, \cdots, k+K-1$, and use the fact that $\mathbb{E}[\ell(W_k)] \leq \mathbb{E}[\ell(\widehat{W}_k)] + C_k$ since $W_k$ is obtained by applying `Opt` to $\widehat{W}_k$, we have

$$\left(\eta_k - \frac{L\eta_k^2}{2}\right)\mathbb{E}[\|P_{\mathcal{S}_k}(G_k)\|^2] - \frac{L\eta_k^2}{2}C - C_k - 2\sum_{i=k+1}^{k+K-1}C_i \leq \mathbb{E}[\ell(W_{k-1}) - \ell(W_{k+K-1})]. \tag{28}$$

By Assumption 3, $\eta_k \to 0$ so without loss of generality, we can assume $\frac{L\eta_k}{2} \leq \frac{1}{2}$ for any $k \in \mathbb{N}$. Therefore for all integer $k \in \mathbb{N}$, we have

$$\frac{\eta_{kK+1}}{2}\mathbb{E}[\|P_{\mathcal{S}_{kK+1}}(G_{kK+1})\|^2] - \frac{L\eta_{kK+1}^2}{2}C - C_{kK+1} - 2\sum_{i=kK+2}^{(k+1)K} C_i \leq \mathbb{E}[\ell(W_{kK}) - \ell(W_{(k+1)K})]. \tag{29}$$

By Assumption 1, there exists a constant $C_\ell$ so that $C_\ell \leq \ell(W)$ for any $W$. Summing up for $k \in \{1, \cdots, T\}$ one has

$$\sum_{k=1}^{T} \frac{\eta_{kK+1}}{2}\mathbb{E}[\|P_{\mathcal{S}_{kK+1}}(G_{kK+1})\|^2] \leq \frac{LC}{2}\sum_{k=1}^{T}\eta_{kK+1}^2 + \sum_{k=1}^{T} C_{kK+1} + 2\sum_{k=1}^{T}\sum_{i=kK+2}^{(k+1)K} C_i \tag{30}$$
$$+ \mathbb{E}[\ell(W_K)] - C_\ell.$$

Taking $T \to \infty$, and note that $\sum_k \eta_k^2 < \infty$ and $\sum_k |C_k| < \infty$, the first and second series on the right hand side of (30) are obviously bounded. For the third series, note that

$$\sum_{k=1}^{\infty}\sum_{i=kK+2}^{(k+1)K} C_i \leq \sum_{k=1}^{\infty}\sum_{i=kK+2}^{(k+1)K} |C_i| \leq \sum_{i=1}^{\infty} |C_i| < \infty. \tag{31}$$

Finally $\|\mathbb{E}[\ell(W_K)] - C_\ell\|$ is obviously bounded for a fixed $K$, and therefore, one has

$$\sum_{k=1}^{\infty}\eta_{kK+1}\mathbb{E}[\|P_{\mathcal{S}_{kK+1}}(G_{kK+1})\|^2] < \infty. \tag{32}$$

By $\sum_k^{\infty} \eta_k = \infty$ and a contradiction argument, one has $\liminf_{k\to\infty} \mathbb{E}[\|P_{\mathcal{S}_k}(G_k)\|] = 0$. Since $\|G_k\| \leq dist(G_k, \mathcal{S}_k) + \|P_{\mathcal{S}_k}(G_k)\| \leq \delta_k + \|P_{\mathcal{S}_k}(G_k)\| \to \delta$ where $\delta := \lim_{k\to\infty} \delta_k$, the final result follows. $\qquad\square$

### G.3 ADDITIONAL THEORETICAL DISCUSSION

There has been a growing body of work establishing exact convergence guarantees for memory-efficient pre-training methods, e.g., (Chen et al., 2025; He et al., 2024; Robert et al., 2024; Zhang et al., 2025a). These results typically specify concrete subspace constructions (such as randomized projections), under which the bias term depending on $\delta_k$ can be eliminated, provided that the algorithm adopts a carefully tuned hyperparameter schedule, for example a $\beta_1$ or learning rate schedule that depends on the iteration horizon. While such schedules are theoretically valid, they often require problem- or time-dependent tuning, which limits their practicality in standard PEFT settings.

To illustrate the compatibility of our framework with such specific cases, we provide a simple case study in which the gradients are deterministic and the subspace $\mathcal{S}_k^{\text{FG}}$ is chosen as the top-$r$ SVD subspace of $G_k$, as in `PESO-LoRA-R`. Under this setting, we can establish an exact asymptotic convergence result. We first present the following lemma, which bounds the error introduced by the projected gradients.

**Lemma G.3.** *Let $G \in \mathbb{R}^{m \times n}$ have singular value decomposition*

$$G = U\Sigma V^\top, \qquad \Sigma = \text{diag}(\sigma_1, \ldots, \sigma_p), \quad \sigma_1 \geq \cdots \geq \sigma_p \geq 0,$$

*where $p = \min\{m, n\}$. Let*

$$U_r := U[:, 1:r], \quad V_r := V[:, 1:r], \quad \Lambda_r := \text{diag}(\sigma_1, \ldots, \sigma_r),$$

*and define $G_r := U_r\Lambda_r V_r^\top$. Then*

$$\|G - G_r\|^2 \leq \left(1 - \frac{r}{p}\right)\|G\|^2. \tag{33}$$

*Proof.* By the Eckart–Young–Mirsky theorem,

$$\|G - G_r\|^2 = \sum_{i=r+1}^{p} \sigma_i^2, \qquad \|G\|^2 = \sum_{i=1}^{p} \sigma_i^2.$$

Since the singular values are sorted in nonincreasing order, we have

$$\frac{1}{r}\sum_{i=1}^{r}\sigma_i^2 \ \geq \ \frac{1}{p}\sum_{i=1}^{p}\sigma_i^2,$$

which implies

$$\sum_{i=1}^{r}\sigma_i^2 \ \geq \ \frac{r}{p}\sum_{i=1}^{p}\sigma_i^2 = \frac{r}{p}\,\|G\|^2.$$

Therefore,

$$\|G - G_r\|^2 = \|G\|^2 - \sum_{i=1}^{r}\sigma_i^2 \ \leq \ \|G\|^2 - \frac{r}{p}\,\|G\|^2 = \left(1 - \frac{r}{p}\right)\|G\|^2,$$

which proves (33). □

Next, we state the exact convergence result.

**Theorem G.4.** *Suppose Assumption 1 and 6 hold. Assume that full-gradient restarts are enabled, the learning rate is chosen as $\eta_k = \frac{1}{L}$, and the subspace $\mathcal{S}_k^{\mathrm{FG}}$ is selected as the top-$r$ SVD subspace of $G_k$, as in* PESO-LoRA-R*. Then the iterates $\{W_k\}$ generated by Algorithm 1 satisfy $\liminf_{k\to\infty}\|G_k\| = 0$.*

*Proof.* The proof follows the same structure as that of Theorem G.2, with one key modification. Since now $\mathcal{S}_{kK+1}$ is chosen as the top-$r$ SVD subspace of $G_{kK+1}$,

$$\|G_{kK+1}\| = \|G_{kK+1} - P_{\mathcal{S}_{kK+1}}(G_{kK+1}) + P_{\mathcal{S}_{kK+1}}(G_{kK+1})\|$$

$$\leq \sqrt{1 - \frac{r}{p}}\,\|G_{kK+1}\| + \|P_{\mathcal{S}_{kK+1}}(G_{kK+1})\|, \tag{34}$$

where $p := \min\{m, n\}$ and the inequality follows from Lemma G.3 and the triangle inequality. It follows from (18) that

$$\left(1 - \sqrt{1 - \frac{r}{p}}\right)\|G_{kK+1}\| \leq \|P_{\mathcal{S}_{kK+1}}(G_{kK+1})\| \to 0,$$

and therefore $\liminf_{k\to\infty}\|G_k\| = 0$. □

## THE USE OF LARGE LANGUAGE MODELS (LLMS)

LLMs did not play a significant role in the conception of this work. The methodology, problem formulation, and theoretical contributions are entirely original and developed independently by the authors. We made limited use of general-purpose LLM tools (ChatGPT and Gemini) for writing polish and occasional code debugging support. No part of the research ideation, design, or substantive writing relied on LLMs.

