# OpenReview forum: "Parameter-Efficient Subspace Optimization for LLM Fine-Tuning"
_ICLR.cc/2026/Conference — Submitted to ICLR 2026_

### Official Review · Reviewer_a9FV · 2025-10-25

**Soundness:** 2
**Presentation:** 2
**Contribution:** 2
**Rating:** 2
**Confidence:** 4

**Summary:**

This paper proposes PESO, a LoRA-like PEFT algorithm motivated by subspace optimization. PESO alternatively explores new subspaces via low-rank SVD like GaLore, and then exploits the subspace via Adam updates. This paper also provides theoretical convergence proofs and conduct experiments to justify PESO's efficiency.

**Strengths:**

1. PESO achieves better parameter efficiency and performance compared to vanilla LoRA, as illustrated in the experiments.
2. The algorithm is new and a theoretical convergence proof is provided.

**Weaknesses:**

1. In Line 109-111, the authors claim that "The resulting algorithm is, to our knowledge, the first memory-efficient method for LLM training with provable convergence to full-parameter optimality up to small errors, without additional assumptions such as explicit low-rankness of the solution." However, proir works have already established exact convergence rates for memory-efficient LLM training methods with standard or mild assumptions, including GoLore [arXiv:2410.11289] and LDAdam [arXiv:2410.16103], both of which were uploaded to arXiv one year ago. Consequently, given the non-diminishing convergence gap in Theorem 5.1 and the presence of these prior works, I highly disagree with this claim.
2. The assumptions in the convergence analysis are too strong. Specifically, the approximation error $\delta_k$ can diverge if the gradient $G_k$ diverges. The present proofs cannot exclude the case where $\lim_\{k\rightarrow\infty}\delta_k=\lim_\{k\rightarrow\infty}\\|G_k\\|_F=\infty$, and thus I believe Assumption 4 is a strong assumption.
3. I think the improvements of PESO, as compared to the baselines in the experiments, are limited. Other subspace optimization algorithms such as GaLore [arXiv:2403.03507], GoLore [arXiv:2410.11289] , LDAdam [arXiv:2410.16103], Fira [arXiv:2410.01623] and Subtrack++ [arXiv:2502.01586] have similar memory efficiency and much stronger performance than LoRA. It is recommended to at least include some of these strong baselines in the experiments.

**Questions:**

1. See Weakness 1. Can the authors provide more evidence to support the claim?
2. See Weakness 2. Can the authors give more detailed explanation on why Assumption 4 holds?
3. See Weakness 3. Is PESO empirically comparable to, or better than the memory-efficient baselines I mentioned?

---

> ### Author Response · Authors · 2025-11-25
>
> > **Q1:** The claim of being the first memory-efficient method with convergence guarantees appears inaccurate given prior works; additional clarification or evidence is needed.
>
> **A1:** We acknowledge the confusion caused by our original statement. As indicated from the title onward, our main focus is **PEFT**, and the intended claim was within the context of **LoRA based PEFT literature**. Specifically, among works such as LoRA-One and LoRA-GA, our method is the first that both provides a convergence guarantee and is explicitly designed for PEFT, leveraging a LoRA style algorithmic structure. We have rephrased the relevant text to avoid ambiguity.
>
> We also acknowledge that our convergence analysis includes a biased term. This is **intentional**: PESO is designed as a **general framework**, not a single algorithm, and therefore must accommodate a wide range of subspace choices and optimizers. Our goal is to provide a **unified iterative subspace minimization perspective** that applies to many explore and exploit variants, rather than rely on assumptions tailored to a specific method.
>
> Because of this generality, we cannot assume a particular subspace construction or optimizer. To ensure theoretical validity while remaining broadly applicable, we introduce **Assumption 4** (exploration quality) and **Assumption 5** (exploitation progress). These may appear non standard only because prior work typically analyzes **individual algorithms**, whereas we analyze a **framework**. Conceptually, Assumption 4 requires that the subspace is informative enough for exploration, and Assumption 5 requires that the exploitation step makes progress, which is easily satisfied in practice with optimizers such as Adam or SGD. Thus, our assumptions are intentionally weaker than those used in the literature.
>
> Importantly, when the subspaces are instantiated with concrete constructions (e.g., randomized subspaces as in GoLore), the bias term can be removed provided that the algorithm adopts a carefully tuned $\beta_1$ schedule that depends on the iteration horizon. Such schedules are theoretically valid but often require problem- or time-dependent tuning, which limits their practicality in standard PEFT settings. This is consistent with our framework-level analysis, and therefore does not conflict with our results. We have added references clarifying this point in the revised manuscript; see Remark 1–2 of Section 5 and Appendix G.3 for a simple case study achieving exact convergence.
>
> Additionally, LDAdam implicitly assumes that its projection subspace $U_t$ satisfies $||(I-U_tU_t^T)B_t||\leq q_r||B_t||$ for $q_r<1$, which is a **strong** and highly nontrivial requirement in $\ell_2$ geometry, severely restricting the choice of $U_t$. PESO’s Assumption 4 captures this idea in a **much more general and flexible way**, consistent with our framework level goals.
>
> We also acknowledge the comment regarding convergence rates. Since our analysis is intentionally developed for a general framework, without fixing the subspace construction or the optimizer, it is not possible to derive a meaningful convergence rate without imposing much stronger algorithm-specific assumptions. For this reason, we focus on an **asymptotic convergence** analysis, which is the strongest statement that can be made at the framework level. A more detailed rate analysis for a specific instantiation such as PESO-LoRA-R/T requires committing to concrete design choices and is therefore left as future work.
>
>
> > **Q2:** The assumptions in the convergence analysis are too strong.
>
> **A2:** We respectfully disagree. As discussed above, we consider Assumption 4 to be quite mild. Regarding your specific example: in general nonconvex optimization, a standard convergence requirement is to show that $\lim_{k\to\infty}||G_k||=0$ (see e.g. a classical textbook reference [1]). If instead one assumes that $\lim_{k\to\infty}||G_k||=\infty$, this already implies that the method diverges from stationarity, and therefore no algorithm can satisfy convergence under such a condition.
> The gap involving $\delta$ is intentional: we aim to impose the weakest possible assumption that still yields a meaningful convergence guarantee for a general framework. As noted earlier, existing works typically rely on much stronger assumptions in order to close this gap.

---

> ### Author Response · Authors · 2025-11-25
>
> > **Q3:** It is unclear whether PESO is empirically comparable to strong memory-efficient baselines; experimental evidence is requested.
>
>
> **A3:** As noted in the Global comments #1 and #2, we did not originally include comparisons with GaLore and other pre-training methods because our focus is different. Our work targets PEFT and follows the benchmark settings of LoRA variants such as LoRA-One, whereas the methods you mentioned are designed primarily for pre-training. We have now added the corresponding comparisons, and the results show that PEFT-specific methods such as PESO-LoRA and LoRA-One significantly outperform these pre-training–oriented approaches; see below or Table 5 of the revised manuscript. Our results are the average values over 3 runs.
>
> |  | STEM | Social Sciences | Humanities | Others | Average |
> | --- | --- | --- | --- | --- | --- |
> | LoRA | 53.50 | 74.85 | 58.97 | 72.34 | 64.25 |
> | GaLore | 54.50 | 75.11 | 58.59 | 72.03 | 64.43 |
> | Fira | 53.53 | 75.46 | 58.59 | 72.09 | 64.32 |
> | APOLLO /w SVD | 54.73 | 75.46 | 58.72 | 72.68 | 64.76 |
> | APOLLO | 54.37 | 75.86 | 58.18 | 71.69 | 64.35 |
> | PESO-LoRA-R | **56.92** | **76.80** | **60.80** | **73.52** | **66.40** |
>
> [1]  J. Nocedal and S. Wright, Numerical Optimization, Springer New York, 2 ed., 1999.

---

> ### Comment · Reviewer_a9FV · 2025-11-28
> **Thank you for the rebuttal.**
>
> I thank the authors for the detailed rebuttal. Below are my responses point by point:
> 1. Regarding A1, it is acceptable if the authors include a fair statement to avoid ambiguity, but it seems that the authors haven't provided the rephrased statement?
> 2. I may agree that building a convergence analysis for a generalized framework is more difficult than for a specific algorithm, however, if the result becomes too bad, e.g., introducing a non-diminishing bias term, I believe: i) either the proposed framework is too general to include bad algorithms; ii) or the theoretical analysis is not tight. A simple method to show the theoretical tightness is to discuss how the analysis for the generalized framework can reduce to the convergence result of some specific algorithms, which have zero biases.
> 3. Regarding A2, I just don't understand the logic here. I believe the assumption is strong because it does not hold when the algorithm diverges (see my original review). The fact implied by the assumption that "gradient norm should not approach infinity" is something you are supposed to prove, not assume. What I suggest is not using a strong assumption that has already excluded a divergent case, but not proving convergence by assuming the opposite. However, in A2 the authors only demonstrate why they should not prove convergence by assuming divergence. I regard this issue as my biggest concern, and would not raise my rating unless this issue had been appropriately resolved.
> 4. Regarding A3, I appreciate the efforts made by the authors. Though not including the SOTA method Subtrack++, the additional results have already partially addressed my concerns regarding the experiments.
>
> In general, I wish the authors could further address my concern regarding the assumptions, which is my primary concern and has not been successfully addressed in the rebuttal.

---

> > ### Author Response · Authors · 2025-11-28
> >
> > We thank the reviewer for the thoughtful follow-up and appreciate the opportunity to clarify several key points.
> >
> > > **Q1:** Missing the rephrased statement.
> >
> > **A1:** Thank you for the follow-up. We acknowledge that the rephrased statements may not have been clearly visible in the responses, but these revisions were **already incorporated** into the updated PDF and all changes were **highlighted in red**. Specifically:
> >
> > 1. **Abstract (lines 18–20):**
> >    *“Importantly, our framework establishes the convergence in the full-parameter space, resolving a critical gap of LoRA variants where low-rank updates lack such guarantees.”*
> >    This revision (already included in the uploaded PDF) rephrased the original claim of being the first to show convergence and clarifies that the scope is specifically **LoRA variants**.
> >
> > 2. **Section 1, Contributions (lines 109–112):**
> >    *“The resulting algorithm is, to our knowledge, the first method for LLM fine-tuning that combines the practical effectiveness of LoRA-style designs with a provable convergence to full-parameter optimality up to small errors, without requiring additional assumptions such as explicit low-rankness of the solution.”*
> >    This was also rephrased earlier to avoid overclaiming and to specify that the “first convergence result” applies to **methods with LoRA-structured updates**.
> >
> >
> >
> > > **Q2:** Quality of the results.
> >
> > **A2:** We agree with the reviewer’s suggested approach, and this is exactly what we implemented in the previous rebuttal; see the last sentence of A1, paragraph 4. We added a simple case study demonstrating **exact convergence** as a concrete instantiation of our framework. Specifically, we show that **PESO-LoRA-R converges exactly when gradients are deterministic**. This is documented in **Remark 2 (line 525)** and **Appendix G.3**.
> >
> > Importantly, this mirrors the analysis in GoLore [1], which we added to the references after the reviewer helpfully pointed it out: methods such as GaLore do **not** converge in general stochastic settings, but can converge under deterministic gradients or very large batch sizes. Otherwise, one must commit to **specific subspace constructions** and **hyperparameter schedules** (e.g., randomized subspaces and a $\beta_1$ schedule in GoLore) to establish exact convergence.
> >
> > Producing an exact convergence theorem for PESO-LoRA-R/T in the full stochastic setting would likewise require fixing concrete design choices, which lies beyond the scope of our framework-level analysis and is left as future work.

---

> ### Author Response · Authors · 2025-11-28
>
> > **Q3:** Logic of the assumption being strong.
>
> **A3:** We appreciate the reviewer’s concern and would like to further clarify Assumption 4.
>
> First, regarding the reviewer’s logical point: we do **not** agree that Assumption 4 “already excludes the divergent case.” Assumption 4 would be "strong" in the suggested sense only if one can show
>
> $\text{Assumption 4 holds} \Rightarrow \lim_{k\to\infty}||G_k|| < \infty,$
>
> but this implication is **false**. For example, consider if $||G_k||\to\infty$ and we choose $S_k=\mathrm{span}(G_k)$, then $\mathrm{dist}(G_k,S_k)=0$, so we may set $\delta_k=0$ for all $k$. Thus Assumption 4 holds trivially in this case **even when gradients diverge**. The reviewer’s original reasoning effectively enforced $\lim_{k\to\infty} \delta_k = \lim_{k\to\infty} ||G_k||$, but to refute Assumption 4 one must show that divergence of $||G_k||$ forces Assumption 4 to fail **for all valid choices of $\delta_k$**, which is not the case. This is because Assumption 4 is an *existence argument on $\delta_k$* and cannot be invalidated by a single class of constructed sequence.
>
> Next, we clarify why Assumption 4 is **weak**. In classical subspace optimization literature, structural assumptions on subspace quality are standard, and in fact necessary, for convergence (see e.g., [2, Eq. (2.2)] and [3, Eq. (2.1)]). In LLM subspace training literature, this typically appears through
> 1. **specifying a concrete subspace** (defining $\mathcal{S}_k$ as e.g., randomized subspaces in GoLore [1], or top-$r$ SVD subspaces in GaLore [4]), or
> 2. **imposing strong contraction conditions**, such as LDAdam’s [5] requirement $||(I - U_t U_t^\top) B_t|| \le q_r\,||B_t||, q_r<1,$ which is a strict requirement on the projection.
>
> By contrast, Assumption 4 imposes **no structural form** on $\mathcal{S}_k$; it only requires that the subspace approximates the gradient with bounded error $\delta_k$. This is aligned with classical assumptions in [2,3] and is strictly weaker than those used in GaLore or LDAdam: in particular, if $\mathcal{S}_k$ is instantiated as the top-$r$ SVD subspace of $G_k$, then standard SVD approximation guarantees imply Assumption 4 automatically holds under certain noise conditions or low-rankness.
>
> In summary, subspace regularity is **necessary** in all convergence analyses of this type in the known literature; Assumption 4 is a **weak, minimal, framework-level** version of this requirement and does **not** preclude divergence. It simply encodes that the chosen subspaces approximate the gradients well, consistent with standard practice in the subspace optimization literature [2,3,4].
>
> > **Q4**: Missing the comparison with Subtrack++
>
> **A4:** We acknowledge the contributions of Subtrack++ [6] and have added it to our references; see lines 150–151. However, we did not include Subtrack++ in our additional empirical comparison because its PEFT evaluation setting is **fundamentally incompatible** with ours. This reflects a broader distinction between **PEFT-specific literature** (e.g., [7,8]) and **pre-training–oriented work** (e.g., [6]).
>
> Subtrack++ evaluates PEFT performance primarily on **non-generative models** such as RoBERTa and does not include large-scale generative benchmarks. In contrast, our work follows the standard PEFT protocols established in [7], evaluating on **T5-based models**, **Llama-2**, and additionally **Llama-3**, which are more representative for modern PEFT research. For this reason, our additional comparisons follow the setting of Fira [9], which aligns with these PEFT standards.
>
> [1] He et al., 2024. Subspace Optimization for Large Language Models with Convergence Guarantees.
>
> [2] Conn et al., 1996. On Iterated-Subspace Minimization Methods for Nonlinear Optimization.
>
> [3] Zhang, 2025. Scalable Derivative-Free Optimization Algorithms with Low-Dimensional Subspace Techniques.
>
> [4] Zhao et al., 2024. GaLore: Memory-Efficient LLM Training by Gradient Low-Rank Projection.
>
> [5] Robert et al., 2025. LDAdam: Adaptive Optimization from Low-Dimensional Gradient Statistics.
>
> [6] Rajabi et al., 2025. SubTrack++: Gradient Subspace Tracking for Scalable LLM Training.
>
> [7] Zhang et al., 2025. One-Step Full Gradient Suffices for Low-Rank Fine-Tuning, Provably and Efficiently.
>
> [8] Wang et al., 2024. LoRA-Pro: Are Low-Rank Adapters Properly Optimized?
>
> [9] Chen et al., 2024. Fira: Can We Achieve Full-rank Training of LLMs Under Low-rank Constraint?

---

### Official Review · Reviewer_yTKV · 2025-10-29

**Soundness:** 3
**Presentation:** 3
**Contribution:** 3
**Rating:** 6
**Confidence:** 3

**Summary:**

In this paper, the authors have introduced a unifying framework, Parameter-Efficient Subspace Optimization (PESO). This framework may cover many existing methods, such as LoRA, and bridge them with algorithms and the theory of subspace optimization.

**Strengths:**

The strengths of this paper are summarized as follows:

1. It has combined multiple Parameter-Efficient Fine-Tuning (PEFT) methods, such as LoRA, AdaLoRA, and GaLore, using a single mathematical view.

2. Theoretically, it has given the first proof of a full-parameter convergence guarantee for memory memory-efficient fine-tuning method. The convergence guarantee is in the full model weight space.

3. The proposed framework, PESO, is practical. It is a plug and play design and can improve existing PEFT methods with very little modification. This seems to be very impactful in this field.

**Weaknesses:**

The weaknesses of this paper are summarized as follows:

1. The experimental results are based on T5-base and LLaMA-2-7B. It would be better if the authors could consider including more experimental results on more models, such as LLaMA 3, and it would be more interesting to test models on different sizes.

2. The experimental results seem to focus on fine-tuning. It would be better if the authors may consider full pre-training. Also, it primarily compares against LoRA-based baselines. It lacks evaluation or comparison on Galore or Galore variants, such as GoLore [1] and Sara [2].

[1] Yutong He, Pengrui Li, Yipeng Hu, Chuyan Chen, and Kun Yuan. "Subspace optimization for large language models with convergence guarantees." ICML'25.

[2] Haochen Zhang, Junze Yin, Guanchu Wang, Zirui Liu, Tianyi Zhang, Anshumali Shrivastava, Lin Yang, and Vladimir Braverman. "Breaking the Frozen Subspace: Importance Sampling for Low-Rank Optimization in LLM Pretraining". NeurIPS'25.

**Questions:**

Please see the weaknesses.

---

> ### Author Response · Authors · 2025-11-25
>
> > **Q1:** The experimental evaluation should cover more models, including LLaMA 3, and explore a wider range of model sizes.
>
> **A1:** Thanks for the suggestion. Our experiments follow the protocol used in PEFT benchmarks such as LoRA-GA and LoRA-One to ensure fair comparison. We also agree that extending to larger models is important, and we have now conducted experiments on **Llama 3** (see below or Table 4 of the revised manuscript). As shown, our method continues to achieve benchmark-level performance, and the advantages become even more pronounced as model size increases, consistent with the Llama 2 results in the main text.
>
> |  | HumanEval | MMLU | GSM8K |
> | --- | --- | --- | --- |
> | LoRA | 42.47 (2.56) | 63.95 (0.05) | 70.64 (0.53) |
> | LoRA-GA | 44.32 (5.64) | 62.91 (0.08) | 76.67 (0.31) |
> | LoRA-One | 45.32 (1.52) | 64.33 (0.14) | 77.71 (0.17) |
> | PESO-LoRA-R | **47.15 (0.76)** | **64.34 (0.21)** | **77.79 (0.18)** |
>
>
> > **Q2:** Extending the evaluation from fine-tuning to pre-training would strengthen the results. And the comparison is limited to LoRA-based baselines; additional baselines such as GaLore and its variants should be included.
>
> **A2:** As stated in the title and introduction, this paper focuses on **PEFT**, although several components could be extended to pre-training. Our primary goal is to improve PEFT benchmarks from the LoRA family using insights from subspace optimization. Due to page and computational resource constraints, we leave a thorough pre-training investigation to future work. Nevertheless, we have added comparisons with GaLore variants and demonstrate clear benefits of our method within the PEFT regime; see below or Table 5 of the revised manuscript. Our results are the average values over 3 runs.
>
> |  | STEM | Social Sciences | Humanities | Others | Average |
> | --- | --- | --- | --- | --- | --- |
> | LoRA | 53.50 | 74.85 | 58.97 | 72.34 | 64.25 |
> | GaLore | 54.50 | 75.11 | 58.59 | 72.03 | 64.43 |
> | Fira | 53.53 | 75.46 | 58.59 | 72.09 | 64.32 |
> | APOLLO /w SVD | 54.73 | 75.46 | 58.72 | 72.68 | 64.76 |
> | APOLLO | 54.37 | 75.86 | 58.18 | 71.69 | 64.35 |
> | PESO-LoRA-R | **56.92** | **76.80** | **60.80** | **73.52** | **66.40** |

---

> > ### Comment · Reviewer_yTKV · 2025-11-26
> >
> > I acknowledge that I have read the responses from the authors. All of my concerns are addressed. I choose to maintain my original rating.

---

### Official Review · Reviewer_XChR · 2025-11-01

**Soundness:** 2
**Presentation:** 2
**Contribution:** 2
**Rating:** 2
**Confidence:** 4

**Summary:**

This paper proposes Parameter-Efficient Subspace Optimization (PESO) — a unifying framework that connects modern parameter-efficient fine-tuning methods for large language models (LLMs), such as LoRA, with the classical theory of subspace optimization. PESO provides a principled foundation that interprets these methods through an exploration–exploitation trade-off in the subspace, leading to the design of new algorithms that are both memory-efficient and have strong convergence guarantees.

**Strengths:**

- Provides a framework that can cover some existing low-rank fine-tuning approaches
- The paper is well-written in general and easy to follow

**Weaknesses:**

While the paper claims contributions at the conceptual, theoretical, and empirical levels, these contributions appear insufficiently substantiated.

1. **Conceptual novelty**. The subspace minimization perspective is not new. This viewpoint has already been well established in GaLore [A1] and more recently revisited in Randomized Subspace Optimization (RSO) [A2]. In particular, the proposed framework in Equation (3) closely resembles RSO, where a low-rank variable $\xi$ is obtained by solving a subproblem and then added back to the base parameter $W$. The authors are encouraged to clearly articulate the distinctions between their framework and the RSO algorithm.

2. **Theoretical contribution**. The convergence analysis is weak and incomplete. Numerous existing works have provided both exact convergence guarantees and explicit convergence rates, such as RSO [A2], LDAdam [A3], SARA [A4], and RAC-LoRA [A5]. By contrast, the proposed algorithm only achieves convergence to a biased solution dependent on $\delta$, without demonstrating exact convergence and convergence rates. This is a major concern regarding the paper’s theoretical rigor.

3. **Experimental evaluation**. The empirical results are not comprehensive. The paper omits comparisons with recent strong baselines, including LDAdam, SARA, and RAC-LoRA, APPOLO [A6] which have demonstrated strong performance in both pre-training and fine-tuning settings.

4. **Assumptions**. In Lines 127–136, the authors argue that prior works rely on unrealistic assumptions such as $r < m$ or random projections. However, Assumptions 4 and 5 in this paper are themselves non-standard and not commonly adopted in the literature. It is therefore unconvincing to claim that the present assumptions are more natural or milder than those in existing studies.

[A1] GaLore: Memory-Efficient LLM Training by Gradient Low-Rank Projection

[A2] A Memory Efficient Randomized Subspace Optimization Method for Training Large Language Models

[A3] LDAdam: Adaptive Optimization from Low-Dimensional Gradient Statistics

[A4] Breaking the Frozen Subspace: Importance Sampling for Low-Rank Optimization in LLM Pretraining

[A5] Randomized Asymmetric Chain of LoRA: The First Meaningful Theoretical Framework for Low-Rank Adaptation

[A6] APOLLO: SGD-like Memory, AdamW-level Performance

**Questions:**

1. Clearly state the difference from existing subspace optimization methods such as RSO [A2]

2. Establish the exact convergence of the proposed algorithm. Establish the convergence rate of the proposed framework. Compare the rates with existing literature.

3. Conduct experiments with stronger baselines such as LDAdam, SARA, and RAC-LoRA, APPOLO

---

> ### Author Response · Authors · 2025-11-25
>
> > **Q1, Conceptual novelty:**  The subspace minimization perspective is not new. This viewpoint has already been well established in GaLore [A1] and more recently revisited in Randomized Subspace Optimization (RSO) [A2].
>
> **A1:** We agree that subspace techniques have been explored in prior work. However, our contribution is **not** the subspace idea itself, but a **unified optimization based framework**, grounded in classical iterative minimization [A7], for **PEFT**. To our knowledge, such an optimization centered and PEFT specific perspective has not appeared in the literature.
>
> Most existing methods construct subspaces through numerical or randomized routines, but **do not provide an optimization formulation that links subspace choice, iterative minimization, and the structure of PEFT tasks**. PESO fills this gap by (i) framing subspace updates through an *explore and exploit* perspective (see Section 2.1), and (ii) leveraging PEFT specific structures such as intrinsic dimensionality and the implicit regularization of LoRA variants (see PESO-LoRA from Section 3). This viewpoint directly guides how newly developed subspace techniques should be incorporated into practical algorithms.
>
> In particular, our work focuses on **fine tuning**, not pre training. PEFT tasks exhibit strong **intrinsic dimensionality**, and LoRA variants empirically demonstrate **implicit regularization** when exploring these subspaces. PESO provides a principled optimization framework to understand and leverage these structures. As shown in the additional experiments in the table below, our PESO guided LoRA variant yields **consistent and significant improvements** over them.
>
> We note we have already discussed the connection between RSO and GaLore and our framework in Section 2.2. Below, we further clarify how PESO differs from and improves upon them:
>
> **(i) RSO.** RSO uses randomized projections, a specific choice of $\mathcal{M}_k$ in our notation (see Table 1 of Section 2.2), with strong expectation guarantees. However, its subspace design is independent of the **local geometry**, which is crucial for practical fine tuning. PESO, following classical nonlinear optimization principles, explicitly promotes **local information driven exploration**, addressing this limitation.
>
> **(ii) GaLore.** GaLore suffers from practical instability reported in [A8], and only proves convergence with fixed subspaces under strong low rank assumptions. PESO, in contrast, is built upon a more principled subspace optimization design, resolving both the theoretical and empirical limitations of GaLore.

---

> ### Author Response · Authors · 2025-11-25
>
> > **Q2, Theoretical contributions:** The convergence analysis is weak and incomplete. The proposed algorithm only achieves convergence to a biased solution, without demonstrating exact convergence and convergence rates.
>
> **A2:** We acknowledge that our convergence analysis includes a biased term. This is **intentional**:   PESO is designed as a **general framework**, not a theory for a single algorithm, and therefore the analysis must accommodate a wide class of possible subspace choices and optimizers. Our goal is to provide a **unified iterative subspace–minimization perspective** that applies to many explore–exploit variants, rather than impose assumptions tailored to any one method.
>
> Because of this generality, we cannot assume a specific subspace construction or optimization routine. To make the theory workable while remaining widely applicable, we introduce **Assumption 4** (exploration quality) and **Assumption 5** (exploitation progress). These may appear non-standard only because the literature usually analyzes **specific algorithms**, whereas we analyze a **framework**. Conceptually, Assumption 4 simply requires that the chosen subspace is sufficiently informative for exploration, and Assumption 5 requires that the exploitation step makes progress which is easily met in practice when using Adam or SGD.
>
> Importantly, when the subspaces are instantiated with concrete constructions (e.g., randomized subspaces as in RSO, SARA and RAC-LoRA), the bias term can be removed provided that the algorithm adopts a carefully tuned hyperparameter schedule, such as a $\beta_1$ or learning rate schedule that depends on the iteration horizon. Such schedules are theoretically valid but often require problem- or time-dependent tuning, which limits their practicality in standard PEFT settings. This is consistent with our framework-level analysis, and therefore does not conflict with our results. We have added references clarifying this point in the revised manuscript; see Remark 1–2 of Section 5 and Appendix G.3 for a simple case study achieving exact convergence.
>
> Additionally, LDAdam implicitly assumes that its projection subspace $U_t$ has to satisfy $||(I-U_tU_t^T)B_t||\leq q_r||B_t||$ for $q_r<1$, which is a strong and highly nontrivial requirement in $\ell_2$ geometry, restricting allowable choices of $U_t$. PESO’s Assumption 4 is deliberately formulated to capture this idea **in a far more general and flexible way**, consistent with our framework-level ambitions.
>
> We also acknowledge the comment regarding convergence rates. Since our analysis is intentionally developed for a general framework, without fixing the subspace construction or the optimizer, it is not possible to derive a meaningful convergence rate without imposing much stronger algorithm-specific assumptions. For this reason, we focus on an **asymptotic convergence** analysis, which is the strongest statement that can be made at the framework level. A more detailed rate analysis for a specific instantiation such as PESO-LoRA-R/T requires committing to concrete design choices and is therefore left as future work.
>
> > **Q3, Additional experiments:** The empirical results are not comprehensive. The paper omits comparisons with recent strong baselines, including LDAdam, SARA, and RAC-LoRA, APPOLO [A6].
>
> **A3:** As noted in the Global comments #1 and #2, we did not originally include comparisons to these methods because they are designed for **pre-training**, while our framework targets **PEFT**. For completeness, we have now added these comparisons; see below or Table 5 of the revised manuscript. As shown here, our PESO-LoRA variant achieves clearly stronger performance on PEFT tasks. Our results are the average values over 3 runs.
>
> |  | STEM | Social Sciences | Humanities | Others | Average |
> | --- | --- | --- | --- | --- | --- |
> | LoRA | 53.50 | 74.85 | 58.97 | 72.34 | 64.25 |
> | GaLore | 54.50 | 75.11 | 58.59 | 72.03 | 64.43 |
> | Fira | 53.53 | 75.46 | 58.59 | 72.09 | 64.32 |
> | APOLLO /w SVD | 54.73 | 75.46 | 58.72 | 72.68 | 64.76 |
> | APOLLO | 54.37 | 75.86 | 58.18 | 71.69 | 64.35 |
> | PESO-LoRA-R | **56.92** | **76.80** | **60.80** | **73.52** | **66.40** |

---

> ### Author Response · Authors · 2025-11-25
>
> > **Q4, Assumptions:** Assumptions 4 and 5 in this paper are themselves non-standard and not commonly adopted in the literature.
>
> **A4:** As discussed in the second point above, our assumptions may appear non-standard only because we analyze a **general framework** capable of unifying a wide class of subspace methods, whereas the literature usually analyzes **specific algorithms**.
>
> Intuitively, both of our additional assumptions are rather weak, as Assumption 4 simply requires that the chosen subspace is sufficiently informative for exploration (up to small errors), and Assumption 5 requires that the exploitation step makes progress which is easily met by Adam or SGD.
>
> [A7] On Iterated Subspace Minimization Methods for Nonlinear Optimization
>
> [A8] Fira: Can We Achieve Full-rank Training of LLMs under Low-rank Constraint?

---

### Official Review · Reviewer_M2iq · 2025-11-01

**Soundness:** 2
**Presentation:** 2
**Contribution:** 2
**Rating:** 2
**Confidence:** 4

**Summary:**

This paper introduces PESO (Parameter-Efficient Subspace Optimization), a unifying framework for parameter-efficient fine-tuning of LLMs grounded in classical subspace minimization. PESO connects methods like LoRA and GaLore to a principled exploration-exploitation paradigm, for memory-efficient optimization with provable convergence in the full parameter space. The authors instantiate PESO into practical variants, PESO-LoRA-R and PESO-LoRA-T.

**Strengths:**

1. The PESO framework bridges PEFT with classical subspace minimization, offering an exploration–exploitation perspective and a unified Algorithm 1 that generalizes several existing methods.
2. PESO-LoRA-R and PESO-LoRA-T emerge as straightforward, practical special cases directly derived from the framework.
3. The paper presents theoretical guarantees for full-rank convergence under the stated assumptions.
4. The model is empirically evaluated through Llama-2-7B pre-training and multiple benchmark experiments.

**Weaknesses:**

1. Since the core theme of the paper revolves around exploration-exploitation, it would be natural to include targeted ablation studies, particularly examining the effects of restart frequency (K), rank (r), and related parameters.
2. Although the paper positions itself as a unifying framework, it lacks in-depth discussion and comparison with key baselines in this area; notably GaLore [1] and other state-of-the-art methods.
3. (Please correct me if I’m mistaken,) but M appears to be defined inconsistently; once as a projection map and elsewhere as a subspace. The notation would benefit from clearer, more consistent presentation. Additionally, there are minor grammatical issues (e.g., line 38: “Therefore, updating the entire …”).
4. The alignment techniques are central to the proposed algorithm and should be discussed thoroughly in the main text, rather than being deferred to the appendix.
5. The model is evaluated primarily against LoRA variants, but several recent strong baselines, including GaLore [1], APOLLO [2], LDAdam [3], FiRA [4], etc., are missing. Moreover, SubTrack++ [5], which also explores identifying optimal subspaces via geometric insights, appears conceptually related to the exploration phase and warrants discussion.
6. The evaluation results are not fully convincing, as the mentioned baselines in point 5 typically outperform LoRA variants. This raises concerns about whether the proposed algorithms offer substantial improvements or meaningful advantages.
7. The computational efficiency of the proposed methods is not addressed; in particular, time and memory costs should be analyzed, given that SVD operations are often computationally expensive.
8. The proposed variants require clearer exposition in the main text, including detailed explanations and mathematical formulations of the optimizers and steps used in Algorithms 2 and 3. The current presentation includes repetitive content, while several important details are relegated to the appendix.


---
[1] Zhao et al., 2024. GaLore: Memory-Efficient LLM Training by Gradient Low-Rank Projection.

[2] Zhu et al., 2025. APOLLO: SGD-like Memory, AdamW-level Performance.

[3] Robert et al., 2025. LDAdam: Adaptive Optimization from Low-Dimensional Gradient Statistics.

[4] Chen et al., 2024. Fira: Can We Achieve Full-rank Training of LLMs Under Low-rank Constraint?

[5] Rajabi et al., 2025. SubTrack++: Gradient Subspace Tracking for Scalable LLM Training

**Questions:**

Please refer to the weaknesses.

---

> ### Author Response · Authors · 2025-11-25
>
> > **Q1:** The paper lacks ablations on key exploration–exploitation parameters such as restart frequency $K$ and rank $r$.
>
> **A1:** We agree that ablation studies are important and were missing. We have now added ablation results on fine-tuning GSM8K with Llama-3.1-8B (see below or Appendix F.3 of the revised manuscript). As shown in the table, increasing the rank $r$ improves accuracy, while varying the restart frequency $K$ yields relatively stable and robust performance.
> Varying rank $r$: evaluated on GSM8K, $K$=500.
>
> | Rank $r$ | 1 | 2 | 4 | 8 | 16 | 32 | 64 |
> | --- | --- | --- | --- | --- | --- | --- | --- |
> | Acc (%) | 75.28 | 75.97 | 76.88 | 77.56 | 77.71 | 78.01 | 78.39 |
>
> Varying $K$ and number of restarts (once/twice): evaluated on GSM8K, $r$=8.
>
> | $K$ | 250 | 500 | 750 | 1000 | infinity (LoRA-One) |
> | --- | --- | --- | --- | --- | --- |
> | Restart Once | 77.18 | 77.56 | 77.18 | 77.26 | 77.26 |
> | Restart Twice | 76.88 | 77.30 | 77.56 | 76.88 | N/A  |
>
>
> > **Q2:** Although the paper positions itself as a unifying framework, it lacks in-depth discussion and comparison with key baselines in this area.
>
> **A2:** We did compare GaLore and projection-bsed methods and explicitly describe how they fit into PESO in Section 2.2, particularly in Table 1 and the paragraph on *Projected Subspace*. Appendix C further discusses how these methods explore and exploit subspaces with improved memory efficiency, which is crucial for **pre-training**.
>
> As discussed in the Global comments (point #1 and #2), we did not include empirical comparisons with GaLore and other pre-training methods in the original manuscript because our focus is PEFT; however, we have now added these comparisons in the revised version; also see below or Table 5 of the revised manuscript. Our results are the average values over 3 runs.
>
> |  | STEM | Social Sciences | Humanities | Others | Average |
> | --- | --- | --- | --- | --- | --- |
> | LoRA | 53.50 | 74.85 | 58.97 | 72.34 | 64.25 |
> | GaLore | 54.50 | 75.11 | 58.59 | 72.03 | 64.43 |
> | Fira | 53.53 | 75.46 | 58.59 | 72.09 | 64.32 |
> | APOLLO /w SVD | 54.73 | 75.46 | 58.72 | 72.68 | 64.76 |
> | APOLLO | 54.37 | 75.86 | 58.18 | 71.69 | 64.35 |
> | PESO-LoRA-R | **56.92** | **76.80** | **60.80** | **73.52** | **66.40** |
>
>
> > **Q3:** The definition of $\mathcal{M}$ is unclear or inconsistent—sometimes treated as a projection map and elsewhere as a subspace.
>
> **A3:** The definitions are consistent. $\mathcal{M}$ and $\mathcal{M}_k$ are defined as dimension-lifting maps that embed low-dimensional coordinates $\xi$ to the original parameter space (lines 76 and 87). $\mathcal{M}_k$ also implicitly defines the subspace used at iteration $k$, since by construction $\mathcal{S}_k := \lbrace\mathcal{M}_k(\xi) : \xi \in \mathbb{R}^d\rbrace$ (line 178) is the subspace where optimization occurs. Thus, the choice of $\mathcal{M}_k$ determines the subspace itself. We have revised the text in Section 1 (lines 79, 88, 90) to clarify this and avoid potential confusion.
>
> > **Q4:** Alignment steps are central to the method but described only in the appendix; the main text lacks an adequate explanation.
>
> **A4:** We agree that the alignment technique is an essential component. We have therefore added a concise summary highlighting the key aspects most relevant to practical implementation, while keeping the extended discussion in the appendix for completeness; see lines 341-347 of Section 3.1 and Appendix D.
>
> > **Q5:** Experiments mainly compare against LoRA variants and omit strong baselines such as GaLore, APOLLO, LDAdam, FiRA, and SubTrack++.
>
> **A5:** As discussed in the Global comments #1 and #2, these methods are **primarily designed for pre-training**, whereas our framework and algorithms focus on PEFT. For completeness, we have added experimental comparisons; see the table above. As expected, PESO-LoRA consistently achieves significantly better performance on PEFT tasks.
>
> We also thank the reviewer for pointing out SubTrack++, which we had missed. This method offers valuable insights by capturing local information via Grassmannian subspaces and can be viewed as an attractive choice of defining $\mathcal{M}_k$ for subspace exploration, followed by Adam for exploitation. We see this as a promising direction for extending the PESO framework, but in this paper we focus on applying PESO to LoRA variants to achieve state-of-the-art performance in PEFT. We have added this reference in our revised manuscript; see lines 144-151.

---

> ### Author Response · Authors · 2025-11-25
>
> > **Q6:** The evaluation results are not fully convincing, as the mentioned baselines in Q5 typically outperform LoRA variants.
>
> **A6:** We respectfully disagree. While pre-training–oriented methods often outperform **vanilla** LoRA on PEFT, they are rarely evaluated against **state-of-the-art LoRA variants** such as LoRA-One and LoRA-GA. In the newly added comparisons shown above, our method **PESO-LoRA-R** achieves clear improvements when fine-tuning Llama-3-Instruct-8B on Alpaca and evaluating on MMLU. Combined with the original Section 4 results—where PESO-LoRA matches or exceeds strong LoRA variants—our method demonstrates benchmark-level performance in the PEFT setting.
>
> > **Q7:** The computational efficiency of the proposed methods is not addressed, particularly time and memory costs should be analyzed, given that SVD operations are often computationally expensive.
>
> **A7:** We note that computational efficiency was already discussed in Section 4.1. To further improve clarity, we have added a comparison table of memory and computational costs in Appendix F.3 and included a summary table below. PESO-LoRA-R has the **same memory footprint** as LoRA, and its overall computational cost is comparable. The low-rank SVD step is performed **infrequently**, and our ablation study shows that typically only **one or two restarts** (excluding initialization) are needed for competitive performance. Moreover, the SVD step accounts for only **1.9%** of the total computation time in our experiments, indicating that its overhead is negligible.
>
> |Method  | LoRA | PESO-LoRA-R |
> |--- | --- | --- |
> |Time | 5h03min | 5h09min |
> |Memory | 25.7G | 25.7G |
>
> > **Q8:** Essential implementation details are missing or placed in the appendix.
>
> **A8:** Due to page limits, the detailed steps of *AdamW* are omitted since it is a standard subroutine (see APOLLO [2] for similar abbreviations). We acknowledge the importance of completeness and have included the full version in Appendix E.
> We agree that parts of Algorithms 2 and 3 may appear similar, but each algorithm in fact has a **distinct goal**, which justifies separate discussion. They demonstrate the practical potential of PESO by advancing either the *exploration* or *exploitation* component.

---

### Author Response · Authors · 2025-11-25

We sincerely thank all reviewers for their valuable feedback and insightful questions, which have greatly helped us refine the clarity and scope of our work. We have incorporated the corresponding revisions into the manuscript, with all changes highlighted in **red**. Before addressing individual comments, we would like to clarify several common concerns raised across the reviews.
1. **Scope of the paper.** Our work focuses on **Parameter-Efficient Fine-Tuning (PEFT)** rather than pre-training. The frequently mentioned baselines (GaLore, APOLLO, LDAdam, FiRA, etc.) are primarily designed and benchmarked for **pre-training**, and thus fall outside our main target setting.
2. **PEFT benchmark design.** The strongest and most relevant baselines for PEFT are LoRA variants such as **LoRA-One**, **LoRA-Pro**, and **LoRA-GA**. Our experiments follow the exact protocols used in these works, under which our method attains state-of-the-art performance. While pre-training methods can be applied to fine-tuning, they do not exploit PEFT-specific structures and therefore underperform on more advanced PEFT tasks (see additional results below).
3. **Additional comparisons.** Following reviewers’ suggestions, we added comparisons with GaLore, APOLLO, and FiRA on MMLU using Llama-3. As shown below, our PEFT-specialized method **PESO-LoRA-R** consistently and clearly outperforms these pre-training–oriented approaches in the fine-tuning regime. Our results are the average values over 3 runs.

|  | STEM | Social Sciences | Humanities | Others | Average |
| --- | --- | --- | --- | --- | --- |
| LoRA | 53.50 | 74.85 | 58.97 | 72.34 | 64.25 |
| GaLore | 54.50 | 75.11 | 58.59 | 72.03 | 64.43 |
| Fira | 53.53 | 75.46 | 58.59 | 72.09 | 64.32 |
| APOLLO /w SVD | 54.73 | 75.46 | 58.72 | 72.68 | 64.76 |
| APOLLO | 54.37 | 75.86 | 58.18 | 71.69 | 64.35 |
| PESO-LoRA-R | **56.92** | **76.80** | **60.80** | **73.52** | **66.40** |

4. **Theoretical clarification.** We have added corollaries establishing exact asymptotic convergence for a specific PESO variant. Our original theory aimed to cover a general class of subspace optimization methods without fixing the optimizer or subspace construction, requiring weak but non-standard assumptions (Assumptions 4–5) and resulting in a bias term. We have revised the text to better explain this choice and included an exact convergence result when certain algorithmic components are specified.

---

### Meta-Review · Area_Chair_3u5o · 2026-01-19

**Summary:**

This paper introduces a unifying framework for parameter-efficient fine-tuning of LLMs --- Parameter-Efficient Subspace Optimization (PESO) --- grounded in classical subspace minimization. PESO connects methods like LoRA and GaLore in a principled exploration-exploitation paradigm for memory-efficient optimization with provable convergence in the full parameter space. The authors instantiate PESO into practical variants, PESO-LoRA-R and PESO-LoRA-T.

**Reviewer Concerns:**

The key concern is the existence of prior similar frameworks, many of which were not even cited in the paper. A very major revision citing and comparing to previous frameworks (such as RSO) is needed. Many other concerns were raised (e.g., non-standard assumptions), some of which were addressed satisfactorily, but many of which were not addressed sufficiently.

**Reviewer Scores:**

I expect the scores would stay unchanged

---

### Decision · Program_Chairs · 2026-01-26

Reject